

# Heat stored in the Earth system: Where does the energy go?
## The GCOS Earth heat inventory team

Karina von Schuckmann[1], Lijing Cheng[2,28], Matthew D. Palmer[3], James Hansen[4], Caterina Tassone[5], Valentin Aich[5], Susheel Adusumilli[6], Hugo Beltrami[7], Tim Boyer[8], Francisco José Cuesta-Valero[7,27], Damien Desbruyères[9], Catia Domingues[10,11], Almudena García-García[7], Pierre Gentine[12], John Gilson[13], Maximilian Gorfer[14], Leopold Haimberger[15], Masayoshi Ishii[16], Gregory C. Johnson[17], Rachel Killick[3], Brian A. King[10], Gottfried Kirchengast[14], Nicolas Kolodziejczyk[18], John Lyman[17], Ben Marzeion[19], Michael Mayer[15,29], Maeva Monier[20], Didier Paolo Monselesan[21], Sarah Purkey[6], Dean Roemmich[6], Axel Schweiger[22], Sonia I. Seneviratne[23], Andrew Shepherd[24], Donald A. Slater[6], Andrea K. Steiner[14], Fiammetta Straneo[6], Mary-Louise Timmermans[25], Susan E. Wijffels[21,26]

[1]Mercator Ocean International, France
[2]Institute of Atmospheric Physics, Chinese Academy of Sciences, China
[3]Met Office Hadley Centre, UK
[4]Columbia University Earth Institute, USA
[5]WMO/GCOS, Switzerland
[6]Scripps Institution of Oceanography, UCSD, San Diego, CA, USA
[7]Climate & Atmospheric Sciences Institute, St. Francis Xavier University, NS, Canada
[8]NOAA's National Centers for Environmental Information
[9]Ifremer, University of Brest, CNRS, IRD, Laboratoire d'Océanographie Physique et Spatiale, France
[10]National Oceanographic Centre, UK
[11]ARC Centre of Excellence for Climate Extremes, University of Tasmania, Hobart, Tasmania, Australia
[12]Earth and Environmental Engineering in the School of Engineering and Applied Sciences, Columbia University, USA
[13]University of California, USA
[14]Wegener Center for Climate and Global Change and Institute of Physics, University of Graz, Austria
[15]Department of Meteorology and Geophysics, University of Vienna, Austria
[16]Department of Atmosphere, Ocean and Earth System Modeling Research, Meteorological Research Institute, Japan
[17]NOAA, Pacific Marine Environmental Laboratory, USA
[18]University of Brest, CNRS, IRD, Ifremer, Laboratoire d'Océanographie Physique et Spatiale, IUEM, France
[19]Institute of Geography and MARUM-Center for Marine Environmental Sciences, University of Bremen, Germany
[20]CELAD/Mercator Ocean International, France
[21]CSIRO Oceans and Atmosphere, Hobart, Tasmania, Australia
[22]Polar Science Center, Applied Physics Laboratory, University of Washington, Seattle, USA
[23]Institute for Atmospheric and Climate Science, ETH, Switzerland
[24]Center for Polar Observation and Modeling, University of Leeds, UK
[25]Department of Earth and Planetary Sciences, Yale University, New Haven, USA
[26]Woods Hole Oceanographic Institution, Massachusetts, United States
[27]Environmental Sciences Program, Memorial University of Newfoundland, NL, Canada
[28]Center for Ocean Mega-Science, Chinese Academy of Sciences, Qingdao, China, 266071
[29]European Centre for Medium-Range Weather Forecasts, Reading, UK

*Correspondence to*: Karina von Schuckmann (karina.von.schuckmann@mercator-ocean.fr)

## Abstract

48

Human-induced atmospheric composition changes cause a radiative imbalance at the top-of-atmosphere which is driving global warming. This Earth Energy Imbalance (EEI) is the most critical number defining the prospects for continued global warming and climate change. Understanding the heat gain of the Earth system – and particularly how much and where the heat is distributed – is fundamental to understanding how this affects warming ocean, atmosphere, and land; rising surface temperature; sea level; and loss of grounded and floating ice, which are fundamental concerns for society. This study is a Global Climate Observing System (GCOS) concerted international effort to update the Earth heat inventory, and presents an updated assessment of ocean warming estimates, and new and updated estimates of heat gain in the atmosphere, cryosphere and land over the period 1960-2018. The study obtains a consistent long-term Earth system heat gain over the period 1971-2018, with a total heat gain of $358 \pm 37$ ZJ, which is equivalent to a global heating rate of $0.47 \pm 0.1$ W/m$^2$. Over the period 1971-2018 (2010-2018), the majority of heat gain is reported for the global ocean with 89% (90%), with 52% for both periods in the upper 700m depth, 28% (30%) for the 700-2000m depth layer, and 9% (8%) below 2000m depth. Heat gain over land amounts to 6% (5%) over these periods, 4% (3%) is available for the melting of grounded and floating ice, and 1% (2%) for atmospheric warming. Our results also show that EEI is not only continuing, it is increasing: the EEI amounts to $0.87 \pm 0.12$ W/m$^2$ during 2010-2018. Stabilization of climate, the goal of the universally agreed UNFCCC in 1992 and the Paris agreement in 2015, requires that EEI be reduced to approximately zero to achieve Earth's system quasi-equilibrium. The amount of $CO_2$ in the atmosphere would need to be reduced from 410 ppm to 353 ppm to increase heat radiation to space by 0.87 W/m$^2$, bringing Earth back towards energy balance. This simple number, EEI, is the most fundamental metric that the scientific community and public must be aware of, as the measure of how well the world is doing in the task of bringing climate change under control, and we call for an implementation of the EEI into the global stocktake based on best available science. Continued quantification and reduced uncertainties in the Earth heat inventory can be best achieved through the maintenance of the current global climate observing system, its extension into areas of gaps in the sampling, as well as to establish an international framework for concerted multi-disciplinary research of the Earth heat inventory as presented in this study. This Earth heat inventory is published at DKRZ (https://www.dkrz.de/) under the doi: https://doi.org/10.26050/WDCC/GCOS_EHI_EXP_v2 (von Schuckmann et al., 2020).

## Introduction

In the Paris Agreement of the United Nations Framework Convention on Climate Change (UNFCCC), article 7 demands that "Parties should strengthen […] scientific knowledge on climate, including research, systematic observation of the climate system and early warning systems, in a manner that informs climate services and supports decision-making." This request of the UNFCCC expresses the need of climate monitoring based on best available science, which is globally coordinated through the Global Climate Observing System (GCOS). In the current Implementation Plan of GCOS, main observation gaps are addressed and it states that "closing the Earth's energy balance […] through observations remain outstanding scientific issues that require high-quality climate records of Essential Climate Variables (ECVs)." (GCOS, 2016). GCOS is asking the broader scientific community to establish the observational requirements needed to meet the targets defined in the GCOS Implementation Plan, and to identify how climate observations could be enhanced and continued into the future in order to monitor the Earth's cycles and the global energy budget. This study addresses and intends to respond to this request.

The state, variability and change of Earth's climate are to a large extent driven by the energy transfer between the different components of the Earth system (Hansen, 2005; Hansen et al., 2011). Energy flows alter clouds, and weather and internal climate modes can temporarily alter the energy balance on sub-annual to multi-decadal timescales (Palmer & McNeall, 2014; Rhein et al., 2013). The most practical way to monitor climate state, variability and change is to continually assess the energy, mainly in the form of heat, in the Earth system (Hansen et al., 2011). All energy entering or leaving the Earth climate system does so in the form of radiation at the top-of-the-atmosphere (TOA) (Loeb et al., 2012). The difference between incoming solar radiation and outgoing radiation, which is the sum of the reflected shortwave radiation and emitted longwave radiation, determines the net radiative flux at TOA. Changes of this global radiation balance at TOA - the so-called Earth Energy Imbalance (EEI) - determines the temporal evolution of Earth's climate: If the imbalance is positive (i.e. less energy going out than coming in), energy in the form of heat is accumulated in the Earth system resulting in global warming - or cooling if the EEI is negative. The various facets and impacts of observed climate change arise due to the EEI, which thus represents a crucial measure of the rate of climate change (von Schuckmann et al., 2016). The EEI is the portion of the forcing that has not yet been responded to (Hansen, 2005). In other words, warming will continue even if atmospheric greenhouse gas (GHG) amounts are stabilized at today's level, and the EEI defines additional global warming that will occur without further change in forcing (Hansen et al., 2017). The EEI is less subject to decadal variations associated with internal climate variability than global surface temperature and therefore represents a robust measure of the rate of climate change (von Schuckmann et al., 2016; Cheng et al., 2017).

The Earth system responds to an imposed radiative forcing through a number of feedbacks, which operate on various different timescales. Conceptually, the relationships between EEI, radiative forcing, and surface temperature change can be expressed as (Gregory & Andrews, 2016):

$$\Delta N_{TOA} = \Delta F_{ERF} - |\alpha_{FP}| \, \Delta T_S \qquad (1)$$

where $\Delta N_{TOA}$ is Earth's net energy imbalance at the Top of the Atmosphere (TOA, in W m$^{-2}$),
$\Delta F_{ERF}$ is the effective radiative forcing (W m$^{-2}$), $\Delta T_S$ is the global surface temperature anomaly
(K) relative to the equilibrium state, and $\alpha_{FP}$ is the net total feedback parameter (W m$^{-2}$ K$^{-1}$), which
represents the combined effect of the various climate feedbacks. Essentially, $\alpha_{FP}$ in Equ. (1) can
be viewed as a measure of how efficient the system is at restoring radiative equilibrium for a unit
surface temperature rise. Thus, $\Delta N_{TOA}$ represents the difference between the applied radiative
forcing and Earth's radiative response through climate feedbacks associated with surface
temperature rise (e.g. Hansen et al., 2011). Observation-based estimates of $\Delta N_{TOA}$ are therefore
crucial both to our understanding of past climate change and for refining projections of future
climate change (Gregory & Andrews, 2016; Kuhlbrodt & Gregory, 2012). The long atmospheric
lifetime of carbon dioxide means that $\Delta N_{TOA}$, $\Delta F_{ERF}$ and $\Delta T_S$ will remain positive for centuries,
even with substantial reductions in greenhouse gas emissions and lead to substantial committed
sea-level rise (Cheng, Abraham, et al., 2019; Hansen et al., 2017; Nauels et al., 2017; Matthew D
Palmer et al., 2018).
However, this conceptual picture is complicated by the presence of unforced internal variability in
the climate system, which adds substantial noise to the real-world expression of this equation
(Gregory et al., 2020; Marvel et al., 2018; Palmer & McNeall, 2014). For example, at time scales
from interannual to decadal periods, the phase of the El Niño Southern Oscillation contributes to
both positive or negative variations in EEI (Cheng, Trenberth, et al., 2019; Loeb et al., 2018;
Johnson and Birnbaum, 2017; Loeb et al., 2012). At multi-decadal and longer time scales,
systematic changes in ocean circulation can significantly alter the EEI as well (Baggenstos et al.,
150    2019).

Time-scales of the Earth climate response to perturbations of the equilibrium Earth energy balance
at TOA are driven by a combination of climate forcing and the planet's thermal inertia: The Earth
system tries to restore radiative equilibrium through increased thermal radiation to space via the
Planck response, but a number of additional Earth system feedbacks also influence the planetary
radiative response (Lembo et al., 2019; Myhre et al., 2013). Time-scales of warming or cooling of
the climate depend on the imposed radiative forcing, the evolution of climate and Earth system
feedbacks with ocean and cryosphere in particular leading to substantial "thermal inertia" (Clark
et al., 2016; Marshall et al., 2015). Consequently, it requires centuries for Earth's surface
temperature to respond fully to a climate forcing.
Contemporary estimates of the magnitude of the Earth's energy imbalance range between about
0.4-0.9 W m$^{-2}$ (depending on estimate method and period, see also conclusion), and are directly
attributable to increases in carbon dioxide and other greenhouse gases in the atmosphere from
human activities (Ciais et al., 2013, Myhre et al., 2013, Rhein et al., 2013, Hansen et al., 2011).
The estimate obtained from climate models (CMIP6) as presented by (Wild, 2020) amounts to 1.1
$\pm$ 0.8 W m$^{-2}$. Since the period of industrialization, the EEI has become increasingly dominated by
the emissions of radiatively active greenhouse gases, which perturb the planetary radiation budget
and result in a positive EEI. As a consequence, excess heat is accumulated in the Earth system,
which is driving global warming (Hansen et al., 2005; 2011). The majority (about 90%) of this
positive EEI is stored in the ocean (Rhein et al., 2013) and can be estimated through the evaluation
of ocean heat content (OHC, e.g. Abraham et al., 2013). According to previous estimates, a small
proportion (~3%) contributes to the melting of Arctic sea ice and land ice (glaciers, the Greenland
and Antarctic ice sheets). Another 4% goes into heating of the land and atmosphere (Rhein et al.,
175 2013).
Knowing where and how much heat is stored in the different Earth system components from a
positive EEI, and quantifying the Earth heat inventory is of fundamental importance to unravel the
current status of climate change, as well as to better understand and predict its implications, and
to design the optimal observing networks for monitoring the Earth heat inventory. Quantifying this
energy gain is essential for understanding the response of the climate system to radiative forcing,
and hence to reduce uncertainties in climate predictions. The rate of ocean heat gain is a key
component for the quantification of the EEI, and the observed surface warming has been used to
estimate the equilibrium climate sensitivity (e.g. Knutti & Rugenstein, 2015). However, further
insight into the Earth heat inventory, particularly to further unravel on where the heat is going, can
have implications on the understanding of the transient climate responses to climate change, and
consequently reduces uncertainties in climate predictions (Hansen et al., 2011). In this paper, we
focus on the inventory of heat stored in the Earth system. The first four sections will introduce the
current status of estimate of heat storage change in the ocean, atmosphere, land and cryosphere,
respectively. Uncertainties, current achieved accuracy, challenges, and recommendations for
future improved estimates are discussed for each Earth system component, and in the conclusion.
In the last chapter, an update of the Earth heat inventory is established based on the results of
sections 1-4, followed by a conclusion.
**1. Heat stored in the ocean**
The storage of heat in the ocean leads to ocean warming (IPCC, 2019), and is a major contributor
to sea-level rise through thermal expansion (WCRP, 2018). Ocean warming alters ocean
stratification and ocean mixing processes (Bindoff et al., 2019), affects ocean currents (Hoegh-
Guldberg, 2018; Monika Rhein et al., 2018; Yang et al., 2016), impacts tropical cyclones (Hoegh-
Guldberg, 2018; Trenberth et al., 2018; Woollings et al., 2012) and is a major player in ocean
deoxygenation processes (Breitburg et al., 2018) and carbon sequestration into the ocean (Bopp et
al., 2013; Frölicher et al., 2018). Together with ocean acidification and deoxygenation, ocean
warming can lead to dramatic changes in ecosystems, biodiversity, population extinctions, coral
bleaching and infectious disease, as well as redistribution of habitat (García Molinos et al., 2016;

Gattuso et al., 2015; Ramírez et al., 2017). Implications of ocean warming are also widespread across Earth's cryosphere (Jacobs et al., 2002; Mayer et al., 2019; Polyakov et al., 2017; Serreze & Barry, 2011; Shi et al., 2018). Examples include the basal melt of ice shelves (Adusumilli et al., 2019; Pritchard et al., 2012; Wilson et al., 2017) and marine terminating glaciers (Straneo & Cenedese, 2015); the retreat and speedup of outlet glaciers in Greenland (King et al., 2018) and in Antarctica (Shepherd, Fricker, et al., 2018) and of tidewater glaciers in South America and in the High Arctic (Gardner et al., 2013).

Opportunities and challenges in forming OHC estimates depend on the availability of in situ subsurface temperature measurements, particularly for global-scale evaluations. Subsurface ocean temperature measurements before 1900 had been obtained from ship-board instrumentation, culminating in the global-scale Challenger expedition (1873–1876) (Roemmich & Gilson, 2009). From 1900 up to the mid-1960s, subsurface temperature measurements relied on ship-board Nansen-Bottle and mechanical bathythermograph (MBT) instruments (Abraham et al., 2013), only allowing limited global coverage and data quality. The inventions of the conductivity-temperature-depth (CTD) instruments in the mid-50s and the Expendable Bathythermograph Observing (XBT) system about ten years later increased the oceanographic capabilities for widespread and accurate (in the case of the CTD) measurements of in situ subsurface water temperature (Abraham et al., 2013; Goni et al., 2019).

With the implementation of several national and international programs, and the implementation of the moored arrays in the tropical ocean in the 1980s, the Global Ocean Observing System (GOOS, https://www.goosocean.org/) started to grow. Particularly the global World Ocean Circulation program (WOCE) during the 1990s obtained a global baseline survey of the ocean from top-to-bottom (King et al., 2001). However, measurements were still limited to fixed point platforms, major shipping routes and Naval and research vessel cruise tracks, leaving large parts of the ocean under-sampled. In addition, detected instrumental biases in MBTs, XBTs and other instruments pose a further challenge for the global scale OHC estimate (Abraham et al., 2013; Ciais et al., 2013; M. Rhein et al., 2013), but significant progress has been made recently to correct biases and provide high-quality data for climate research (Boyer et al., 2016; Cheng et al., 2016; Goni et al., 2019; Gouretski & Cheng, 2020). Satellite altimeter measurements of sea surface height began in 1992, and are used to complement in situ derived OHC estimates, either for validation purposes (Cabanes et al., 2013), or to complement the development of global gridded ocean temperature fields (Guinehut et al., 2012; Willis et al., 2004). Indirect estimates of OHC from remote sensing through the global sea level budget became possible with satellite-derived ocean mass information in 2002 (Dieng et al., 2017; Llovel et al., 2014; Loeb et al., 2012; Meyssignac et al., 2019; von Schuckmann et al., 2014).

After the Oceanobs conference in 1999, the international Argo profiling float program was launched with first Argo float deployments in the same year (Riser et al., 2016; Roemmich &

Gilson, 2009). By the end of 2006, Argo sampling had reached its initial target of data sampling
roughly every 3 degrees between 60°S-60°N. However, due to technical evolution, only 40% of
Argo floats provided measurements down to 2000 m depth in the year 2005, but that percentage
increased to 60% in 2010 (von Schuckmann & Le Traon, 2011). The starting point of Argo-based
'best estimate' for near-global-scale (60°S-60°N) OHC is either defined in 2005 (von Schuckmann
and Le Traon, 2011), or in 2006 (Wijffels et al., 2016). The opportunity for improved OHC
estimation provided by Argo is tremendous, and has led to major advancements in climate science,
particularly on the discussion of the EEI (Hansen et al., 2011; Johnson, et al., 2018; Loeb et al.,
2012; Trenberth & Fasullo, 2010; von Schuckmann et al., 2016, Meyssignac et al., 2019). The
near-global coverage of the Argo network also provides an excellent test bed for the long-term
OHC reconstruction extending back well before the Argo period (Cheng et al., 2017). Moreover,
these evaluations inform further observing system recommendations for global climate studies, i.e.
gaps in the deep ocean layers below 2000m depth, in marginal seas, in shelf areas and in the polar
regions (e.g. von Schuckmann et al., 2016), and their implementations are underway, for example
for deep Argo (Johnson et al., 2019).
Different research groups have developed gridded products of subsurface temperature fields for
the global ocean using statistical models (Gaillard et al., 2016; Good et al., 2013; Ishii et al., 2017;
Levitus et al., 2012) or combined observations with additional statistics from climate models
(Cheng et al., 2017). An exhaustive list of the pre-Argo products can be found in for example
Abraham et al., 2013; Boyer et al., 2016; WCRP, 2018; Meyssignac et al., 2019. Additionally,
specific Argo-based products are listed on the Argo webpage (http://www.argo.ucsd.edu/).
Although all products rely more or less on the same database, near-global OHC estimates show
some discrepancies which result from the different statistical treatments of data gaps, the choice
of the climatology and the approach used to account for the MBT and XBT instrumental biases
(Boyer et al., 2016; Wang et al., 2018). Argo-based products show smaller differences, likely
resulting from different treatments of currently under-sampled regions (e.g. von Schuckmann et
al., 2016). Ocean reanalysis systems have been also used to deliver estimates of near-global OHC
(Meyssignac et al., 2019; von Schuckmann et al., 2018), and their international assessments show
increased discrepancies with decreasing in situ data availability for the assimilation (Palmer et al.,
2017; Storto et al., 2018). Climate models have also been used to study global and regional ocean
heat changes and the associated mechanisms, with observational datasets providing valuable
benchmarks for model evaluation (Cheng et al., 2016; Gleckler et al., 2016).
International near-global OHC assessments have been performed previously (e.g. Abraham et al.,
2013; Boyer et al., 2016; Meyssignac et al., 2019; WCRP, 2018). These assessments are
challenging, as most of the gridded temperature fields are research products, and only few are
distributed and regularly updated operationally (e.g. https://marine.copernicus.eu/). This initiative
relies on the availability of data products, their temporal extensions, and direct interactions with
the different research groups. A complete view of all international temperature products can be
only achieved through a concerted international effort, and over time. In this study, we do not
achieve a holistic view of all available products, but present a starting point for future international
regular assessments of near-global OHC. For the first time, we propose an international ensemble
mean and standard deviation of near-global OHC (Fig. 1) which is then used to build an Earth
climate system energy inventory (section 5). The ensemble spread gives an indication of the
agreement among products and can be used as a proxy for uncertainty. The basic assumption for
the error distribution is Gaussian with a mean of zero, which can be approximated by an ensemble
of various products. However, it does not account for systematic errors that may result in biases
across the ensemble and does not represent the full uncertainty. The uncertainty can also be
estimated in other ways including some purely statistical methods (Levitus et al., 2012) or methods
explicitly accounting for the error sources (Lyman & Johnson, 2013), but each method has its
caveats, for example the error covariances are mostly unknown, so adopting a straightforward
method with a "data democracy" strategy has been chosen here as a starting point.
However, future evolution of this initiative is needed to include missing and updated in situ-based
products, ocean reanalyses, as well as indirect estimates (for example satellite-based). The
continuity of this activity will help to further unravel uncertainties due to the community's
collective efforts on detecting/reducing errors, and then provides up-to-date scientific knowledge
of ocean heat uptake.

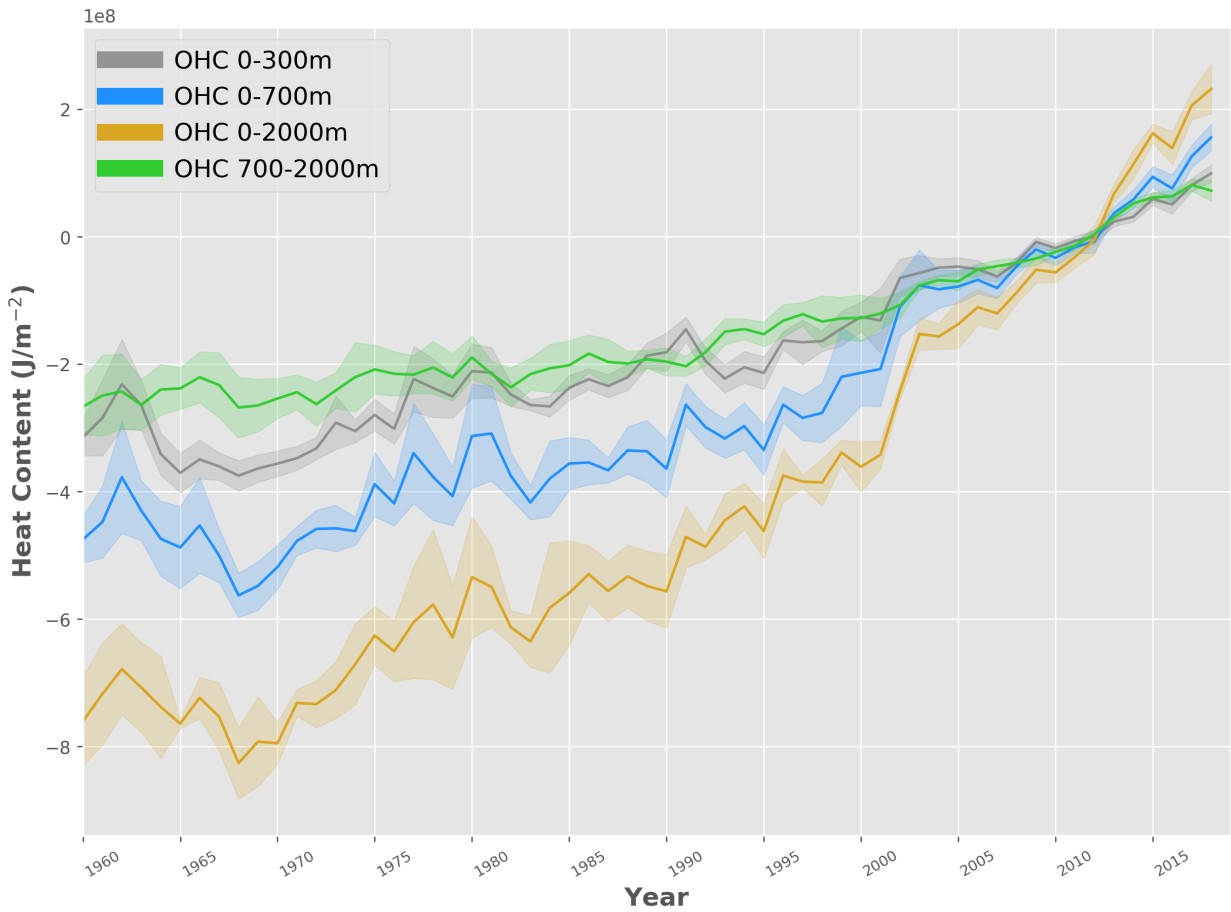

**Figure 1:** *Ensemble mean time series and ensemble standard deviation (2-sigma, shaded) of global ocean*
*heat content (OHC) anomalies relative to the 2005-2017 climatology for the 0-300m (grey), 0-700m (blue),*
*0-2000m (yellow) and 700-2000m depth layer (green). The ensemble mean is an outcome of an*
*international assessment initiative, and all products used are referenced in the legend of Fig. 2. The trends*
*derived from the time series are given in Table 1. Note that values are given for the ocean surface area*
*between 60°S-60°N, and limited to the 300m bathymetry of each product, respectively.*

| Period | 0-300m (W/m²) | 0-700m (W/m²) | 0-2000m (W/m²) | 700-2000m (W/m²) |
|--------|---------------|---------------|----------------|------------------|
| 1960-2018 | 0.3 ± 0.03 | 0.4 ± 0.1 | 0.5 ± 0.1 | 0.2 ± 0.03 |
| 1993-2018 | 0.4 ± 0.04 | 0.6 ± 0.1 | 0.9 ± 0.1 | 0.3 ± 0.03 |
| 2005-2018 | 0.4 ± 0.1 | 0.6 ± 0.1 | 1.0 ± 0.2 | 0.4 ± 0.1 |
| 2010-2018 | 0.5 ± 0.1 | 0.7 ± 0.1 | 1.3 ± 0.3 | 0.5 ± 0.1 |

*Table 1: Linear trends (weighted least square fit, see for example von Schuckmann and Le Traon, 2011) as*
*derived from the ensemble mean as presented in Fig. 1 for different time intervals, and different integration*
*depth. The uncertainty on the trend estimate is given for the 95% confidence level. Note that values are*
*given for the ocean surface area between 60°S-60°N, and limited to the 300m bathymetry of each product,*
*respectively. See text and Fig. 1 caption for more details on the OHC estimates.*


Products used for this assessment are referenced in the caption of Fig. 2. Estimates of OHC have
been provided by the different research groups under homogeneous criteria. All estimates use a
coherent ocean volume limited by the 300m isobath of each product, and are limited to 60°S-60°N
since most observational products exclude high latitude ocean areas because of the low
observational coverage, and only annual averages have been used. 60°S-60°N constitutes ~91%
of the global ocean surface area and limiting to 300m isobath neglects the contributions from
coastal and shallow waters, so the resultant OHC trends will be underestimated if these ocean
regions are warming. For example, neglecting shallow waters can account for 5-10% for 0-2000m
OHC trends (von Schuckmann et al., 2014). A first initial test using Cheng et al., (2017) data
indicates that OHC 0-2000m trends can be underestimated by ~10% if the ocean warming in the
area polewards of 60° latitude is not taken into account (not shown). This is a caveat of the
assessment in this review and will be addressed in the future.

The assessment is based on three distinct periods to account for the evolution of the observing
system, i.e. 1960-2018 (i.e. 'historical'), 1993-2018 (i.e. 'altimeter era') and 2005-2018 (i.e.
'golden Argo-era'). In addition, ocean warming rates over the past decade are specifically
discussed according to an apparent acceleration of global surface warming since 2010 (WMO,
2020; Blunden and Arndt, 2019). All time series reach the end in 2018 – which was one of the
principal limitations for the inclusion of some products. Our final estimates of OHC for the upper
2000m over different periods are the ensemble average of all products, with the uncertainty range
defined by the standard deviation (2-sigma) of the corresponding estimates used (Fig. 1).

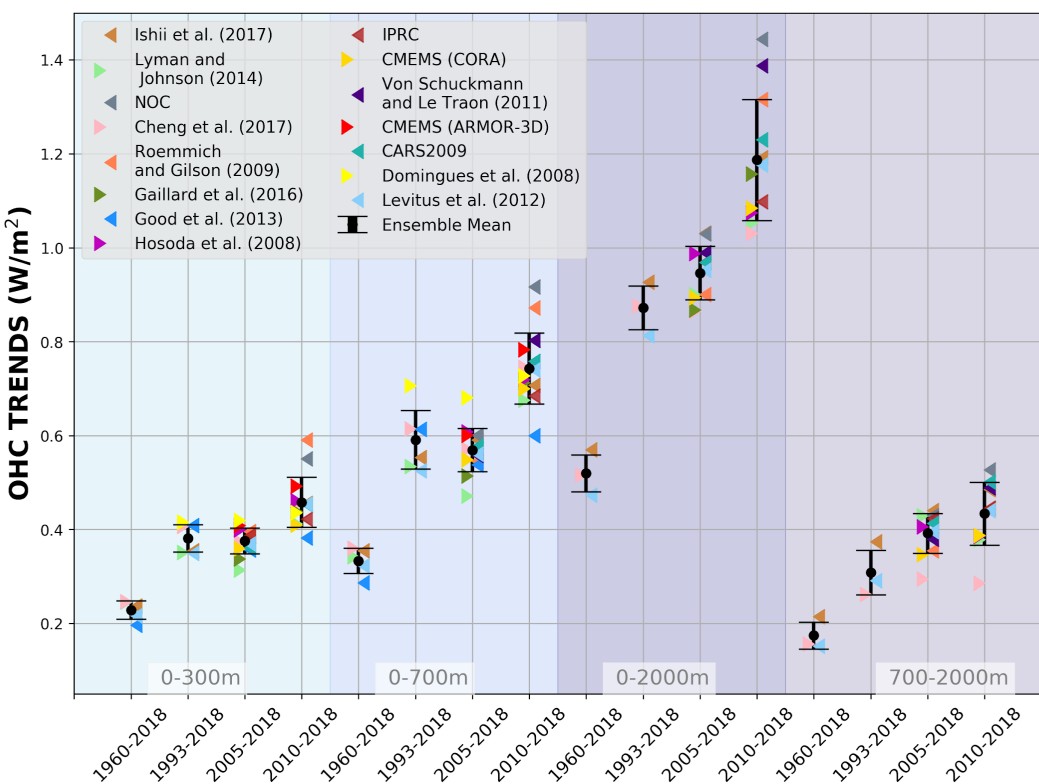

***Figure 2:*** *Linear trends of global ocean heat content (OHC) as derived from different temperature products*
*(colors).* *References* *are* *given* *in* *the* *figure* *legend,* *except* *for* *IPRC*
*(http://apdrc.soest.hawaii.edu/projects/Argo/),* *CMEMS* *(CORA* *&* *ARMOR-3D,*
*http://marine.copernicus.eu/science-learning/ocean-monitoring-indicators),* *CARS2009*
*(http://www.marine.csiro.au/~dunn/cars2009/) and NOC (National Oceanographic Institution,* *(D. G.*
*Desbruyères et al., 2016). The ensemble mean and standard deviation (2-sigma) is given in black,*
*respectively. The shaded areas show trends from different depth layer integrations, i.e. 0-300m (light*
*turquoise), 0-700m (light blue), 0-2000m (purple) and 700-2000m (light purple). For each integration*
*depth layer, trends are evaluated over the three study periods, i.e. historical (1960-2018), altimeter era*
*(1993-2018) and golden Argo era (2005-2018). In addition, the most recent period 2010-2018 is included.*
*See text for more details on the international assessment criteria. Note that values are given for the ocean*
*surface area (see text for more details).*


The first and principal result of the assessment (Fig. 1) is an overall increase of the trend for the
more recent two study periods e.g., the altimeter era (1993-2018) and golden Argo era (2005-2018)
relative to the historical era (1960-2018), which is in agreement with previous results (e.g.
Abraham et al., 2013). The trend values are all given in Table 1. A major part of heat is stored in
the upper layers of the ocean (0-300m and 0-700m depth). However, heat storage at intermediate
depth (700-2000m) increases at a comparable rate as reported for the 0-300m depth layer (Table

1, Fig. 2). There is a general agreement among the 15 international OHC estimates (Fig. 2). However, for some periods and depth layers the standard deviation reaches maximal values up to about 0.3 W/m$^2$. All products agree on the fact that ocean warming rates have increased in the past decades, and doubled since the beginning of the altimeter era (1993-2018 compared with 1960-2018) (Fig. 2). Moreover, there is a clear indication that heat sequestration into the deeper ocean layers below 700m depth took place over the past 6 decades linked to an increase of OHC trends over time (Fig. 2). In agreement with observed accelerated Earth surface warming over the past decade (WMO, 2020; Blunden and Arndt, 2019), ocean warming rates for the 0-2000m depth layer also reached record rates of 1.3 (0.9) ± 0.3 W/m$^2$ for the ocean (global) area over the period 2010-2018.

For the deep OHC changes below 2000m, we adapted an updated estimate from (Purkey & Johnson, 2010) (PG10) from 1991 to 2018, which is a constant linear trend estimate (1.15 +/- 0.57 ZJ/year, 0.07 ± 0.04 W/m$^2$). Some recent studies strengthened the results in PG10 (D. G. Desbruyères et al., 2016; Zanna et al., 2019). Desbruyères et al., (2016) examined the decadal change of the deep and abyssal OHC trends below 2000m in 1990s and 2000s, suggesting that there has not been a significant change in the rate of decadal global deep/abyssal warming from the 1990's to the 2000's and the overall deep ocean warming rate is consistent with PG10. Using a Green Function method, Zanna et al. (2019) reported a deep ocean warming rate of ~0.06 Wm$^{-2}$ during the 2000s, consistent with PG10 used in this study. Zanna et al. (2019) shows a fairly weak global trend during the 1990s, inconsistent with observation-based estimates. This mismatch might come from the simplified or misrepresentation of surface-deep connections using ECCO reanalysis data and the use of time-mean Green's functions in Zanna et al. (2019), as well as from the limited spatial resolution of the observational network for relatively short time-spans. Furthermore, combining hydrographic and deep-Argo floats, a recent study (Johnson et al., 2019) reported an accelerated warming in the South Pacific Ocean in recent years, but a global estimate of the OHC rate of change over time is not available yet.

Before 1990, we assume zero OHC trend below 2000m, following the methodology in IPCC-AR5 (Rhein et al., 2013). The zero-trend assumption is made mainly because there are too few observations before 1990 to make an estimate of OHC change below 2000m. But it is a reasonable assumption because OHC 700-2000m warming was fairly weak before 1990 and heat might not have penetrated down to 2000m (Cheng et al., 2017). Zanna et al. (2019) also shows a near zero OHC trend below 2000m from the 1960s to 1980s. The derived time series is used for the Earth energy inventory in section 5. A centralized (around the year 2006) uncertainty approach has been applied for the deep (> 2000m depth) OHC estimate following the method of Cheng et al., (2017), which allows to extract an uncertainty range over the period 1993-2018 within the given [lower (1.15 -0.57 ZJ/year), upper (1.15 + 0.57 ZJ/year)] range of the deep OHC trend estimate. We then extend the obtained uncertainty estimate back from 1993 to 1960, with 0 OHC anomaly.

## 2. Heat available to warm the atmosphere

While the amount of heat accumulated in the atmosphere is small compared to the ocean, warming of the Earth's near-surface air and atmosphere aloft is a very prominent effect of climate change, which directly affects society. Atmospheric observations clearly reveal a warming of the troposphere over the last decades (Santer et al., 2017; Steiner et al., 2020) and changes in the seasonal cycle (Santer et al., 2018). Changes in atmospheric circulation (Cohen et al., 2014; Fu et al., 2019) together with thermodynamic changes (Fischer & Knutti, 2016; Trenberth et al., 2015) will lead to more extreme weather events and increase high impact risks for society (Coumou et al., 2018; Zscheischler et al., 2018). Therefore, a rigorous assessment of the atmospheric heat content in context with all Earth's climate subsystems is important for a full view on the changing climate system.

The atmosphere transports vast amounts of energy laterally and strong vertical heat fluxes occur at the atmosphere's lower boundary. The pronounced energy and mass exchanges within the atmosphere and with all other climate components is a fundamental element of Earth's climate (Peixoto & Oort, 1992). In contrast, long-term heat accumulation in the atmosphere is limited by its small heat capacity as the gaseous component of the Earth system (von Schuckmann et al., 2016).

Recent work revealed inconsistencies in earlier formulations of the atmospheric energy budget (Mayer et al., 2017; Trenberth & Fasullo, 2018), and hence a short discussion of the updated formulation is provided here. In a globally averaged and vertically integrated sense, heat accumulation in the atmosphere arises from a small imbalance between net energy fluxes at the top of the atmosphere (TOA) and the surface (denoted s). The heat budget of the vertically integrated and globally averaged atmosphere (indicated by the global averaging operator $\langle . \rangle$) reads as follows (Mayer et al., 2017):

$$\langle \frac{\partial AE}{\partial t} \rangle = \langle N_{TOA} \rangle - \langle F_s \rangle - \langle F_{snow} \rangle - \langle F_{PE} \rangle, \qquad (2)$$

where, in mean-sea-level altitude ($z$) coordinates used here for integrating over observational data, the vertically integrated atmospheric energy content $AE$ per unit surface area [Jm$^{-2}$] reads

$$AE = \int_{z_s}^{z_{TOA}} \rho (c_v T + g(z - z_s) + L_e q + \frac{1}{2}V^2)\, dz. \qquad (3)$$

In Equation 2, $AE$ represents the total atmospheric energy content, $N_{TOA}$ the net radiation at top-of-the atmosphere, $F_s$ net surface energy flux defined as the sum of net surface radiation and latent and sensible heat flux, and $F_{snow}$ the latent heat flux associated with snowfall (computed as the product of latent heat of fusion and snowfall rate). Here, we take constant latent heat of vaporization (at 0°C) in the latent heat flux term that is contained in $F_s$ but variations in latent heat flux arising from the deviation of evaporated water from 0°C are contained in $F_{PE}$, which additionally accounts for sensible heat of precipitation (referenced to 0°C). That is, $F_{PE}$ expresses

a modification of $F_s$ arising from global evaporation and precipitation occurring at temperatures
different from 0°C.
Snowfall is the fraction of precipitation that returns originally evaporated water to the surface in a
frozen state. In that sense, $F_{snow}$ represents a heat transfer from the surface to the atmosphere: it
warms the atmosphere through additional latent heat release (associated with freezing of vapor)
and snowfall consequently arrives at the surface in an energetic state lowered by this latent heat.
This energetic effect is most obvious over the open ocean, where falling snow requires the same
amount of latent heat to be melted again and thus cools the ocean. Over high latitudes, $F_{snow}$ can
attain values up to 5 Wm$^{-2}$, but its global average value is smaller than 1 Wm$^{-2}$ (Mayer et al., 2017).
Although its global mean energetic effect is relatively small, it is systematic and should be included
for accurate diagnostics. Moreover, snowfall is an important contributor to the heat and mass
budget of ice sheets and sea ice (see section 4).
$F_{PE}$ represents the net heat flux arising from the different temperatures of rain and evaporated
water. This flux can be sizable regionally, but it is small in a global average sense (warming of the
atmosphere ~0.3 Wm$^{-2}$ according to Mayer et al., 2017).
Equation 3 provides a decomposition of the atmospheric energy content $AE$ into sensible heat
energy (sum of the first two terms, internal heat energy and gravity potential energy), latent heat
energy (third term) and kinetic energy (fourth term), where $\rho$ is the air density, $c_v$ the specific heat
for moist air at constant volume, $T$ the air temperature, $g$ the acceleration of gravity, $L_e$ the
temperature-dependent effective latent heat of condensation (and vaporization) $L_v$ or sublimation
$L_s$ (the latter relevant below 0 °C), $q$ the specific humidity of the moist air, and $V$ the wind speed.
We neglect atmospheric liquid water droplets and ice particles as separate species, as their amounts
and especially their trends are small.
In the $AE$ derivation from observational datasets based on Equation 3, we accounted for the
intrinsic temperature dependence of the latent heat of water vapor by assigning $L_e$ to $L_v$ if ambient
temperatures are above 0 °C and to $L_s$ (adding in the latent heat of fusion $L_f$) if they are below –10
°C, respectively, with a gradual (half-sine weighted) transition over the temperature range
between. The reanalysis evaluations similarly approximated $L_e$ by using values of $L_v$, $L_s$, and $L_f$,
though in slightly differing forms. The resulting differences in AE anomalies from any of these
choices are negligibly small, however, since the latent heat contribution at low temperatures is
itself very small.
As another small difference, the $AE$ estimations from observations neglected the kinetic energy
term in Equation 3 (fourth term), while the reanalysis evaluations accounted for it. This as well
leads to negligible $AE$ anomaly differences, however, since the kinetic energy content and trends
at global scale are more than three orders of magnitude smaller than for the sensible heat (Peixoto
and Oort, 1992). Aligning with the terminology of ocean heat content (OHC) and given the
dominance of the heat-related terms in Equation 3, we hence refer to the energy content $AE$ as
atmospheric heat content (AHC) hereafter.
Turning to the actual datasets used, atmospheric energy accumulation can be quantified using
various data types, as summarized in the following. Atmospheric reanalyses combine
observational information from various sources (radiosondes, satellites, weather stations, etc.) and
a dynamical model in a statistically optimal way. This data type has reached a high level of
maturity, thanks to continuous development work since the early 1990s (Hersbach et al., 2018).
Especially reanalysed atmospheric state quantities like temperature, winds, and moisture are
considered to be of high quality and suitable for climate studies, although temporal discontinuities
introduced from the ever-changing observation system remain a matter of concern (Berrisford et
al., 2011; Chiodo & Haimberger, 2010).
Here we use the current generation of atmospheric reanalyses as represented by ECMWF's fifth-
generation reanalysis ERA5 (Hersbach et al., 2018, 2020), NASA's Modern-Era Retrospective
analysis for Research and Applications version 2 (MERRA2) (Gelaro et al., 2017), and JMA's 55-
year-long reanalysis JRA55 (Kobayashi et al., 2015). All these are available over 1980 to 2018
(ERA5 also in 1979) while JRA55 is the only one covering the full early timeframe 1960 to 1979.
We additionally used a different version of JRA55 that assimilates only conventional observations
also over the satellite era from 1979 onwards, which away from the surface only leaves radiosondes
as data source and which is available to 2012 (JRA55C). The advantage of this product is that it
avoids potential spurious jumps associated with satellite changes. Moreover, JRA55C is fully
independent of satellite-derived Global Positioning System (GPS) radio occultation (RO) data that
are also separately used and described below together with the observational techniques.
In addition to these four reanalyses, the datasets from three different observation techniques have
been used for complementary observational estimates of the atmospheric heat content (AHC). We
use the Wegener Center (WEGC) multi-satellite RO data record, WEGC OPSv5.6 (Angerer et al.,
2017) as well as its radiosonde (RS) data record derived from the high-quality Vaisala sondes
RS80/RS92/VS41, WEGC Vaisala (Ladstädter et al., 2015). WEGC OPSv5.6 and WEGC Vaisala
provide thermodynamic upper air profiles of air temperature, specific humidity, and density from
which we locally estimate the vertical AHC based on the first three integral terms of Equation 3
(Kirchengast et al., 2019). In atmospheric domains not fully covered by the data (e.g., in the lower
part of the boundary layer for RO or over the polar latitudes for RS) the profiles are vertically
completed by collocated ERA5 information. The local vertical AHC results are then averaged into
regional monthly means, which are finally geographically aggregated to global AHC. Applying
this estimation approach in the same way to reanalysis profiles sub-sampled at the observation
locations accurately leads to the same AHC anomaly time series records as the direct estimation
from the full gridded fields.
The third observation-based AHC dataset derives from a rather approximate estimation approach
using the microwave sounding unit (MSU) data records (Mears & Wentz, 2017). Because the very
coarse vertical resolution of the brightness temperature measurements from MSU does not enable
integration according to Equation 3, this dataset is derived by replicating the method used in IPCC
AR5 WGI Assessment Report 2013 (Rhein, M., et al., 2013; Chap. 3, Box 3.1 therein). We used
the most recent MSU Remote Sensing System (RSS) V4.0 temperature dataset (Mears and Wentz,
2017), however, instead of MSU RSS V3.3 that was used in the IPCC AR5 (Mears & Wentz,
2009a, 2009b) (updated to version 3.3). In order to derive global time series of AHC anomalies,
the approach simply combines weighted MSU lower tropospheric temperature and lower
stratospheric temperature changes (TLT and TLS channels) converted to sensible heat content
changes via global atmospheric mass, and an assumed fractional increase of latent heat content
according to water vapor content increase driven by temperature at a near-Clausius-Clapeyron rate
(7.5 %/°C).
Figure 3 shows the resulting global AHC change inventory over 1980 to 2018 in terms of AHC
anomalies of all data types (top), mean anomalies and time-average uncertainty estimates including
long-term AHC trend estimates (middle), and annual-mean AHC tendency estimates (bottom). The
mean anomaly time series (middle left), preceded by the small JRA55 anomalies over 1960-1979
is used as part of the overall heat inventory in Section 5 below. Results including MSU in addition
are separately shown (right column), since this dataset derives from a fairly approximate
estimation as summarized above and hence is given lower confidence than the others deriving from
rigorous AHC integration & aggregation. Since it was the only dataset for AHC change estimation
in the IPCC AR5 report, bringing it into context is considered relevant, however.
The results clearly show that the AHC trends have intensified from the earlier decades represented
by the 1980-2010 trends of near 1.8 TW (consistent with the trend interval used in the IPCC AR5
report). We find the trends about 2.5 times higher over 1993-2018 (about 4.5 TW) and about three
times higher in the most recent two decades over 2002-2018 (near 5.3 TW), a period that is already
fully covered also by the RO and RS records (which estimate around 6 TW). Checking the
sensitivity of these long-term trend estimates to ENSO interannual variations, by comparing trend
fits to ENSO-corrected AHC anomalies (with ENSO regressed out via the Nino 3.4 Index),
confirms that the estimates are robust.
The year-to-year annual-mean tendencies in AHC, reaching amplitudes as high as 50 to 100 TW
(or 0.1 to 0.2 $Wm^{-2}$, if normalized to the global surface area), indicate the strong coupling of the
atmosphere with the uppermost ocean. This is mainly caused by the ENSO interannual variations
that lead to net energy changes in the climate system including the atmosphere (Loeb et al., 2012;
Mayer et al., 2013) and substantial reshuffling of heat energy between the atmosphere and the
upper ocean (Lijing Cheng, Trenberth, et al., 2019; Johnson & Birnbaum, 2017; Mayer et al., 2014,
553  2016).

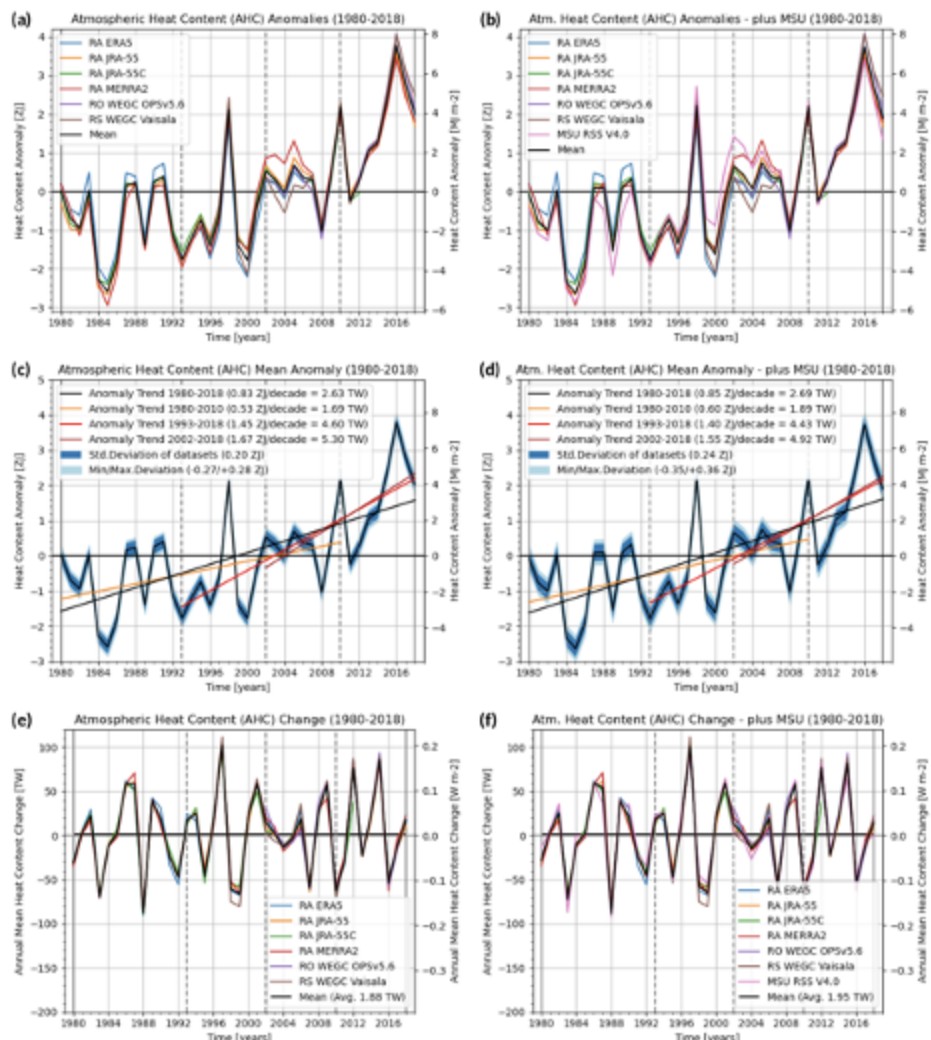


*Figure 3: Annual-mean global AHC anomalies over 1980 to 2018 of four different reanalyses and two (left) or three (right, plus MSU) different observational datasets shown together with their mean (top), the mean AHC anomaly shown together with four representative AHC trends and ensemble spread measures of its underlying datasets (middle), and the annual-mean AHC change (annual tendency) shown for each year over 1980 to 2018 for all datasets and their mean (bottom). The in-panel legends identify the individual datasets shown (top and bottom) and the chosen trend periods together with the associated trend values and spread measures (middle), the latter including the time-average standard deviation and minimum/maximum deviations of the individual datasets from the mean.*

## 3. Heat available to warm land

Although the land component of the Earth's energy budget accounts for a small proportion of heat in comparison with the ocean, several land-based processes sensitive to the magnitude of the available land heat play a crucial role in the future evolution of climate. Among others, the stability

and extent of the continental areas occupied by permafrost soils depend on the land component. Alterations of the thermal conditions at these locations have the potential to release long-term stored $CO_2$ and $CH_4$, and may also destabilize the recalcitrant soil carbon (Bailey et al., 2019; Hicks Pries et al., 2017). Both of these processes are potential "tipping points" (Lenton et al., 2019; Lenton, 2011; Lenton et al., 2008) leading to possible positive feedback on the climate system (Leifeld et al., 2019; MacDougall et al., 2012). Increased land energy is related to decreases in soil moisture that may enhance the occurrence of extreme temperature events (Jeong et al., 2016; S. Seneviratne et al., 2006, 2014; S. I. Seneviratne et al., 2010; Xu et al., 2019). Such extreme events carry negative health effects for the most vulnerable sectors of human and animal populations and ecosystems (Matthews et al., 2017; McPherson et al., 2017; Sherwood & Huber, 2010; Watts et al., 2019). Given the importance of properly determining the fraction of EEI flowing into the land component, recent works have examined the CMIP5 simulations and revealed that Earth System Models (ESMs) have shortcomings in modelling the land heat content of the last half of the 20th century (Francisco José Cuesta-Valero et al., 2016). Numerical experiments have pointed to an insufficient depth of the Land Surface Models (LSMs) (MacDougall et al., 2008, 2010; C. W. Stevens, 2007) and to a zero heat-flow bottom boundary condition (BBC) as the origin of the limitations in these simulations. An LSM of insufficient depth limits the amount of energy that can be stored in the subsurface. The zero heat-flow BBC neglects the small but persistent long-term contribution from the flow of heat from the interior of the Earth, that shifts the thermal regime of the subsurface towards or away from the freezing point of water, such that the latent heat component is misrepresented in the northern latitudes (Hermoso de Mendoza et al., 2018). Although the heat from the interior of the Earth is constant at time scales of a few millennia, it may conflict with the setting of the LSM initial conditions in ESM simulations. Modelling experiments have also allowed to estimate the heat content in land water reservoirs (Vanderkelen et al., 2020), accounting for 0.3±0.3 ZJ from 1900 to 2020. Nevertheless, this estimate has not been included here because it is derived from model simulations and its magnitude is small in relation to the rest of components of the Earth's heat inventory.

**Borehole Climatology**

The main premise of borehole climatology is that the subsurface thermal regime is determined by the balance of the heat flowing from the interior of the Earth (the bottom boundary condition) and the heat flowing through the interface between the lower atmosphere and the ground (the upper boundary condition). If the thermal properties of the subsurface are known, or if they can be assumed constant over short-depth intervals, then the thermal regime of the subsurface can be determined by the physics of heat diffusion. The simplest analogy is the temperature distribution along a (infinitely wide) cylinder with known thermal properties and constant temperature at both ends. If upper and lower boundary conditions remain constant (i.e. internal heat flow is constant, and there are no persistent variations on the ground surface energy balance), then the thermal

regime of the subsurface is well known and it is in a (quasi) steady state. However, any change to
the ground surface energy balance would create a transient, and such a change in the upper
boundary condition would propagate into the ground leading to changes in the thermal regime of
the subsurface (Beltrami, 2002a). These changes in the ground surface energy balance propagate
into the subsurface and are recorded as departures from the quasi-steady thermal state of the
subsurface. Borehole climatology uses these subsurface temperature anomalies to reconstruct the
ground surface temperature changes that may have been responsible for creating the subsurface
temperature anomalies we observe. That is, it is an attempt to reconstruct the temporal evolution
of the upper boundary condition. Ground Surface Temperature Histories (GSTHs) and Ground
Heat Flux Histories (GHFHs) have been reconstructed from borehole temperature profile (BTP)
measurements at regional and larger scales for decadal and millennial time-scales (Barkaoui et al.,
2013; Beck, 1977; Beltrami, 2001; Beltrami et al., 2006; Beltrami & Bourlon, 2004; Cermak,
1971; Christian Chouinard & Mareschal, 2009; Davis et al., 2010; Demezhko & Gornostaeva,
2015; Harris & Chapman, 2001; Hartmann & Rath, 2005; Hopcroft et al., 2007; Huang et al., 2000;
Jaume-Santero et al., 2016; Lachenbruch & Marshall, 1986; Lane, 1923; Pickler et al., 2018; Roy
et al., 2002; Vasseur et al., 1983). These reconstructions have provided independent records for
the evaluation of the evolution of the climate system well before the existence of meteorological
records. Because subsurface temperatures are a direct measure, which unlike proxy reconstructions
of past climate do not need to be calibrated with the meteorological records, they provide an
independent way of assessing changes in climate. Such records, are useful tools for evaluating
climate simulations prior to the observational period (Beltrami et al., 2017; Cuesta-Valero et al.,
2019; Cuesta-Valero et al., 2016; García-García et al., 2016; González-Rouco et al., 2006; Jaume-
Santero et al., 2016; MacDougall et al., 2010; Stevens et al., 2008), as well as for assessing proxy
data reconstructions (Beltrami et al., 2017; Jaume-Santero et al., 2016).
Borehole reconstructions have, however, certain limitations. Due to the nature of heat diffusion,
temperature changes propagated through the subsurface suffer both a phase shift and an amplitude
attenuation (Smerdon & Stieglitz, 2006). Although subsurface temperatures continuously record
all changes in the ground surface energy balance, heat diffusion filters out the high frequency
variations of the surface signal with depth, thus the annual cycle is detectable up to approximately
16 m of depth, while millennial changes are recorded approximately to a depth of 500 m.
Therefore, reconstructions from borehole temperature profiles represent changes at decadal to
millennial time scales. Additionally, borehole data are sparse, since the logs were usually recorded
from holes of opportunity at mining exploration sites. As a result, the majority of profiles were
measured in the Northern Hemisphere, although recent efforts have been taken to increase the
sampling rate in South America (Pickler et al., 2018) and Australia (Suman et al., 2017). Despite
this uneven sampling, the spatial distribution of borehole profiles has been able to represent the
evolution of land surface conditions at global scales (Beltrami & Bourlon, 2004; Cuesta-Valero et
al., 2020; González-Rouco et al., 2006, 2009; Pollack & Smerdon, 2004). Another factor that
reduces the number of borehole profiles suitable for climate analyses is the presence of non-
climatic signals in the measured profiles, mainly caused by groundwater flow and changes in the
lithology of the subsurface. Therefore, all profiles are screened before the analysis in order to
remove questionable logs. Despite all these limitations, the borehole methodology has been shown
to be reliable based on observational analyses (Bense & Kooi, 2004; C Chouinard & Mareschal,
2007; Pollack & Smerdon, 2004; Verdoya et al., 2007) and pseudo-proxy experiments
(García Molinos et al., 2016; González-Rouco et al., 2006, 2009).
**Land Heat Content Estimates**
Global continental energy content has been previously estimated from geothermal data retrieved
from a set of quality-controlled borehole temperature profiles. Ground heat content was estimated
from heat flux histories derived from BTP data (Beltrami, 2002b; Beltrami et al., 2002, 2006).
Such results have formed part of the estimate used in AR3, AR4 and AR5 IPCC reports (see Box
3.1, Chapter 3 (Rhein et al., 2013). A continental heat content estimate was inferred from
meteorological observations of surface air temperature since the beginning of the 20th century
(Huang, 2006). Nevertheless, all global estimates were performed nearly two decades ago. Since,
those days, advances in borehole methodological techniques (Beltrami et al., 2015; Cuesta-Valero
et al., 2016; Jaume-Santero et al., 2016), the availability of additional BTP measurements, and the
possibility of assessing the continental heat fluxes in the context of the FluxNet measurements
(Gentine et al., 2020) requires a comprehensive summary of all global ground heat fluxes and
continental heat content estimates.

| Reference | Time period | Heat Flux (mWm$^{-2}$) | Heat Content (ZJ) | Source of Data |
|---|---|---|---|---|
| (Beltrami, 2002b) | 1950-2000 | 33 | 7.1 | Geothermal |
| Beltrami et al. (2002) | 1950-2000 | 39.1 (3.5) | 9.1 (0.8) | Geothermal |
| Beltrami et al. (2002) | 1900-2000 | 34.1 (3.4) | 15.9 (1.6) | Geothermal |
| (Beltrami, 2002b) | 1765-2000 | 20.0 (2.0) | 25.7 (2.6) | Geothermal |
| Huang (2006) | 1950-2000 | - | 6.7 | Meteorological |
| Gentine et al (2020) | 2004-2015 | 240 (120) | - | FluxNet, Geothermal, LSM |
| Cuesta-Valero et al (2020) | 1950-2000 | 70 (20) | 16 (3) | Geothermal |
| Cuesta-Valero et al (2020) | 1993-2018 | 129 (28) | 14 (3) | Geothermal |
| Cuesta-Valero et al., (2020) | 2004-2015 | 136 (28) | 6 (1) | Geothermal |

*Table 2. Ground surface heat flux and global continental heat content. Uncertainties in parenthesis.*


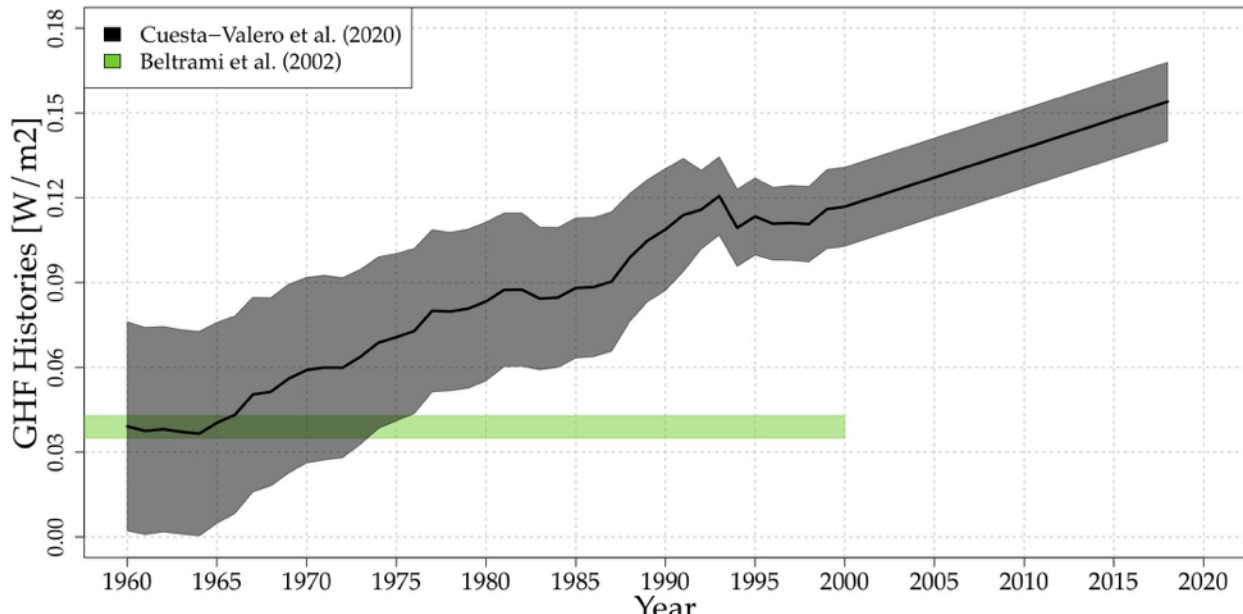


*Figure 4:* *Global mean ground heat flux history (black line) and 95% confidence interval (gray shadow)*
*from BTP measurements from Cuesta-Valero et al. (2020). Results for 1950-2000 from Beltrami et al.*
*(2002) (green bar) are provided for comparison purposes.*


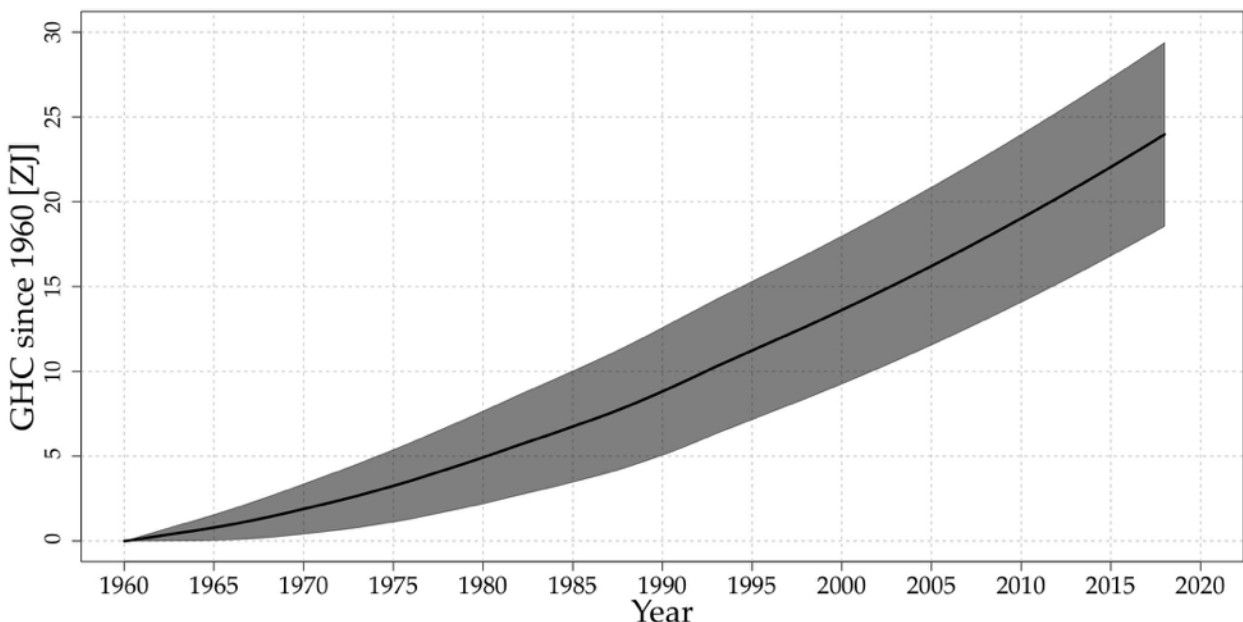


*Figure 5:* *Global cumulative heat storage within continental landmasses since 1960 CE (black line) and*
*95% confidence interval (gray shadow) estimated from GHF results displayed in Fig.4. Data obtained from*
*Cuesta-Valero et al. (2020).*

The first estimates of continental heat content used borehole temperature versus depth profile data. However, the dataset in those analyses included borehole temperature profiles of a wide range of depths, as well as different data acquisition dates. That is, each borehole profile contained the record of the accumulation of heat in the subsurface for different time intervals. In addition, the borehole data were analyzed for a single ground surface temperature model using a single constant value for each of the subsurface thermal properties.

Although the thermal signals are attenuated with depth, which may partially compensate for data shortcomings, uncertainties were introduced in previous analyses that may have affected the estimates of subsurface heat change. A continental heat content change estimate was carried out using a gridded meteorological product of surface air temperature by (Huang, 2006). Such work yielded similar values as the estimates from geothermal data (see Table 2). This estimate, however, assumed that surface air and ground temperatures are perfectly coupled everywhere, and used a single value for the thermal conductivity of the ground. Studies have shown that the coupling of the surface air and ground temperatures is mediated by several processes that may influence the ground surface energy balance, and therefore, the air-ground temperature coupling (García-García et al., 2019; Melo-Aguilar et al., 2018; Stieglitz & Smerdon, 2007). In a novel attempt to reconcile continental heat content from soil heat-plate data from the FluxNet network with estimates from geothermal data and a deep bottom-boundary land surface model simulation, (Gentine et al., 2020) obtained a much larger magnitude from the global land heat flux than all previous estimates. Cuesta-Valero et al. (2020) has recently updated the estimate of the global continental heat content using a larger borehole temperature database (1079 logs) that includes more recent measurements and a stricter data quality control. The updated estimate of continental heat content change also takes into account the differences in borehole logging time and restricts the data to the same depth range for each borehole temperature profile. Such depth range restriction ensures that the subsurface accumulation of heat at all BTP sites is synchronous. In addition to the standard method for reconstructing heat fluxes with a single constant value for each subsurface thermal property, Cuesta-Valero et al. (2020) also developed a new approach that considers a range of possible subsurface thermal properties, several models, each at a range of resolutions yielding a more realistic range of uncertainties for the fraction of the EEI flowing into the land subsurface.

Global land heat content estimates from FluxNet data, geothermal data and model simulations point to a marked increase in the amount of energy flowing into the ground in the last few decades (Fig. 4, 5 and Table 2). These results are consistent with the observations of ocean, cryosphere and atmospheric heat storage increases during the same time period and with EEI at the top of the atmosphere.

## 4. Heat utilized to melt ice

The energy uptake by the cryosphere is given by the sum of the energy uptake within each one of its components: sea-ice, the Greenland and Antarctic ice sheets, glaciers other than those that are part of the ice sheets ('glaciers', hereafter), snow and permafrost. The basis for the heat uptake by

the cryosphere presented here is provided by a recent estimate for the period 1979 to 2017 (F.
Straneo et al., 2020). This study concludes that heat uptake over this period is dominated by the
mass loss from Arctic sea-ice, glaciers and the Greenland and Antarctic ice sheets. The
contributions from thawing permafrost and shrinking snow cover are either negligible, compared
to these other components, or highly uncertain. (Note that warming of the land in regions where
permafrost is present is accounted for in the land-warming, however, the energy to thaw the
permafrost is not). Antarctic sea-ice shows no explicit trend over the period described here
(Parkinson, 2019).  Here, we extend the estimate of Straneo et al. 2020 backwards in time to 1960
and summarize the method, the data and model outputs used. The reader is referred to Straneo et
al. (2020) for further details.
Within each component of the cryosphere, energy uptake is dominated by that associated with
melting; including both the latent heat uptake and the warming of the ice to its freezing point. As
a result, the energy uptake by each component is directly proportional to its mass loss (Straneo et
al., 2020). For consistency with previous estimates (Ciais et al., 2013), we use a constant latent
heat of fusion of $3.34 \times 10^5$ J/kg, a specific heat capacity of $2.01 \times 10^3$ J/kg C and an ice density of
920 kg/m$^3$.
For Antarctica, we separate contributions from grounded ice loss and floating ice loss building on
recent separate estimates for each. Grounded ice loss from 1992 to 2017 is based on a recent study
that reconciles mass balance estimates from gravimetry, altimetry and input-output methods from
1992 to 2017 (Shepherd, Ivins, et al., 2018). For the 1972-1991 period, we used estimates from
(Rignot et al., 2019), which combined modeled surface mass balance with ice discharge estimates
from the input/output method. Floating ice loss between 1994 and 2017 is based on thinning rates
and iceberg calving fluxes estimated using new satellite altimetry reconstructions (Adusumilli et
al., 2019). For the 1960–1994 period, we also considered mass loss from declines in Antarctic
Peninsula ice shelf extent (Cook & Vaughan, 2010) using the methodology described in Straneo
et al. (2020).
To estimate grounded ice mass loss in Greenland, we use the Ice Sheet Mass Balance
Intercomparison Exercise for the time period 1992-2017 (Shepherd et al., 2020) and the difference
between surface mass balance and ice discharge for the period 1979-1991 (Mankoff et al., 2019;
Mouginot et al., 2019; Noël et al., 2018).  Due to a lack of observations, from 1960-1978 we
assume no mass loss. For floating ice mass change, we collated reports of ice shelf thinning and/or
collapse together with observed tidewater glacier retreat (Straneo et al., 2020). Based on firn
modeling we assessed that warming of Greenland's firn has not yet contributed significantly to its
energy uptake (Ligtenberg et al., 2018; F. Straneo et al., 2020).
For glaciers we combine estimates for glaciers from the Randolph Glacier Inventory outside of
Greenland and Antarctica, based on direct and geodetic measurements (Zemp et al., 2019), with

estimates based on a glacier model forced with an ensemble of reanalysis data (Marzeion et al., 2015) and GRACE based estimates (Bamber et al., 2018). An additional contribution from uncharted glaciers or glaciers that have already disappeared is obtained from (Parkes & Marzeion, 2018). Greenland and Antarctic peripheral glaciers are derived from Zemp et al., (2019) and Marzeion et al. (2015).

Finally, while estimates of Arctic sea-ice extent exist over the satellite record, sea-ice thickness distribution measurements are scarce making it challenging to estimate volume changes. Instead we use the Pan-Arctic Ice Ocean Modeling and Assimilation System (PIOMAS) (Schweiger et al., 2011; Zhang & Rothrock, 2003) which assimilates ice concentration and sea surface temperature data and is validated with most available thickness data (from submarines, oceanographic moorings, and remote sensing; and against multi-decadal records constructed from satellite, e.g. (Labe et al., 2018; Laxon et al., 2013; X. Wang et al., 2016). A longer reconstruction using a slightly different model version, PIOMAS-20C (Schweiger et al., 2019), is used to cover the 1960 to 1978 period that is not covered by PIOMAS.

These reconstructions reveal that all four components contributed similar amounts (between 2-5 ZJ) over the 1960-2017 period amounting to a total energy uptake by the cryosphere of 14.7 +/- 1.9 ZJ. Compared to earlier estimates, and in particular the 8.83 ZJ estimate from (Ciais et al., 2013), this larger estimate is a result both of the longer period of time considered and, also, the improved estimates of ice loss across all components, especially the ice shelves in Antarctica. Approximately half of the Cryosphere's energy uptake is associated with the melting of grounded ice, while the remaining half is associated with the melting of floating ice (ice shelves in Antarctica and Greenland, Arctic sea-ice).

## 5. The Earth heat inventory: Where does the energy go?

The Earth has been in radiative imbalance, with less energy exiting the top of the atmosphere than entering, since at least about 1970 and the Earth has gained substantial energy over the past 4 decades (Hansen, 2005; Rhein et al., 2013). Due to the characteristics of the Earth system components, the ocean with its large mass and high heat capacity dominates the Earth heat inventory (Cheng et al., 2016, 2017; Rhein et al., 2013; von Schuckmann et al., 2016). The rest goes into grounded and floating ice melt, and warming the land and atmosphere.


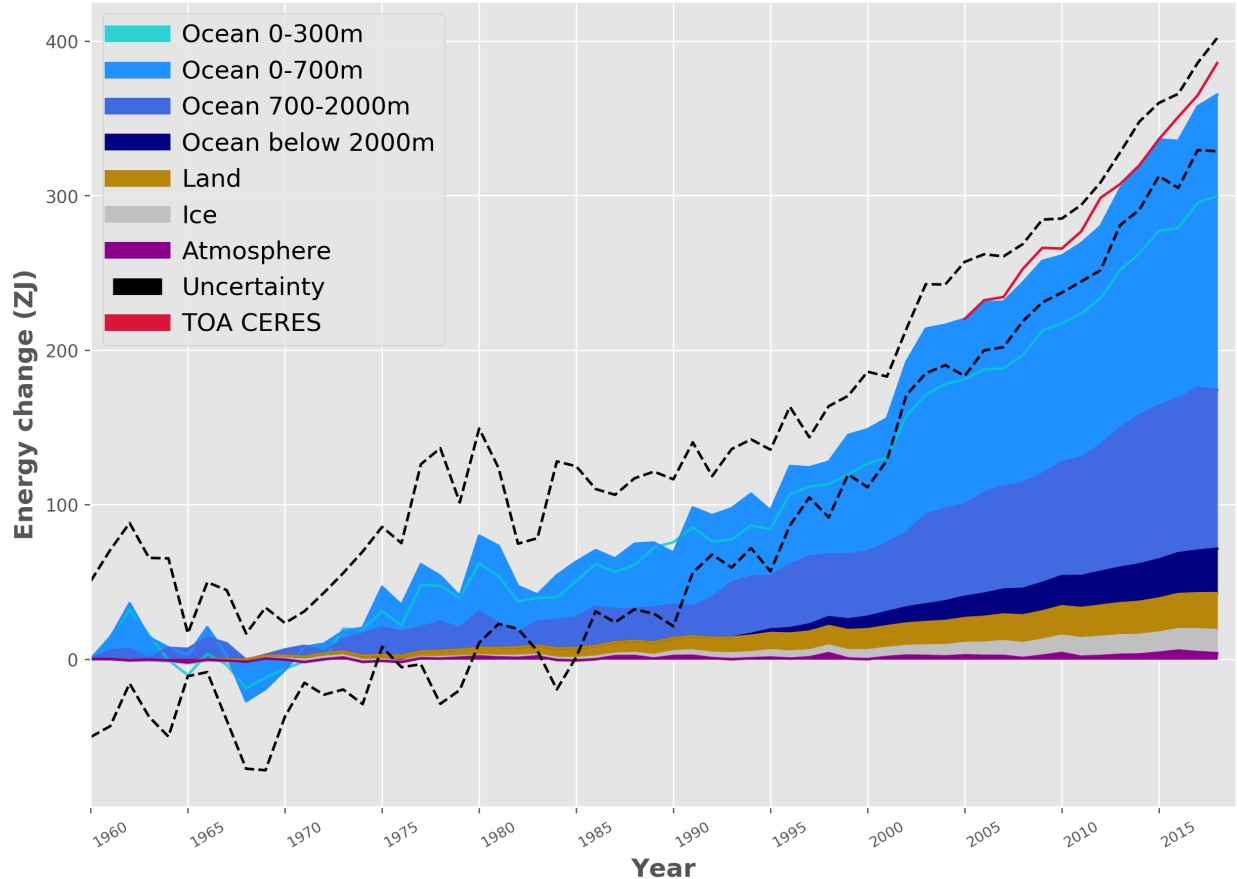

***Figure 6:*** *Earth heat inventory (energy accumulation) in ZJ (1 ZJ = $10^{21}$ J) for the components of the*
*Earth's climate system relative to 1960 and from 1960 to 2018 (assuming constant cryosphere increase for*
*the year 2018). See section 1-4 for data sources. The upper ocean (0-300m, light blue line, and 0-700m,*
*light blue shading) account for the largest amount of heat gain, together with the intermediate ocean (700-*
*2000m, blue shading), and the deep ocean below 2000m depth (dark blue shading). Although much lower,*
*the second largest contributor is the storage of heat on land (orange shading), followed by the gain of heat*
*to melt grounded and floating ice in the cryosphere (gray shading). Due to its low heat capacity, the*
*atmosphere (magenta shading) makes a smaller contribution. Uncertainty in the ocean estimate also*
*dominates the total uncertainty (dot-dashed lines derived from the standard deviations (2-sigma) for the*
*ocean, cryosphere and land. Atmospheric uncertainty is comparably small). Deep ocean (> 2000m) are*
*assumed zero before 1990 (see section 1 for more details). The dataset for the Earth heat inventory is*
*published at DKRZ (https://www.dkrz.de/) under the doi:*
*https://doi.org/10.26050/WDCC/GCOS_EHI_EXP_v2. The net flux at TOA from the NASA CERES*
*program is shown in red (https://ceres.larc.nasa.gov/data/, see also for example Loeb et al., 2012) for the*
*period 2005-2018 to account for the 'golden period' of best available estimates. We obtain a total heat*
*gain of 358 ± 37 ZJ over the period 1971-2018, which is equivalent to a heating rate (i.e. the EEI) of 0.47*
*± 0.1 $Wm^{-2}$ applied continuously over the surface area of the Earth (5.10 × $10^{14}$ $m^2$). The corresponding*
*EEI over the period 2010-2018 amounts to 0.87 ± 0.12 $Wm^{-2}$. A weighted least square fit has been used*
*taking into account the uncertainty range (see also von Schuckmann and Le Traon, 2011).*


In agreement with previous studies, the Earth heat inventory based on most recent estimates of
heat gain in the ocean (section 1), the atmosphere (section 2), land (section 3) and the cryosphere
(section 4) shows a consistent long-term heat gain since the 1960s (Fig. 6). Our results show a total
heat gain of $358 \pm 37$ ZJ over the period 1971-2018, which is equivalent to a heating rate of 0.47
$\pm 0.1$ Wm$^{-2}$, and applied continuously over the surface area of the Earth ($5.10 \times 10^{14}$ m$^2$). For
comparison, the heat gain obtained in IPCC AR5 amounts to $274 \pm 78$ ZJ and 0.4 Wm$^{-2}$ over the
period 1971-2010 (Rhein et al., 2013). In other words, our results show that since the IPCC AR5
estimate has been performed, heat accumulation has continued at a comparable rate. The major
player in the Earth inventory is the ocean, particularly the upper (0-700m) and intermediate (700-
2000m) ocean layers (see also section 1, Fig. 2).

Although the net flux at TOA as derived from remote sensing are anchored by an estimate of global
OHC (Loeb et al., 2012), and thus does not provide a completely independent result for the total
EEI, we additionally compare net flux at TOA with the Earth heat inventory obtained in this study
(Fig. 6). Both rate of changes compare well, and we obtain $0.7 \pm 0.1$ Wm$^{-2}$ for the remote sensing
estimate at TOA, and $0.8 \pm 0.1$ Wm$^{-2}$ for the Earth heat inventory over the period 2005-2018.

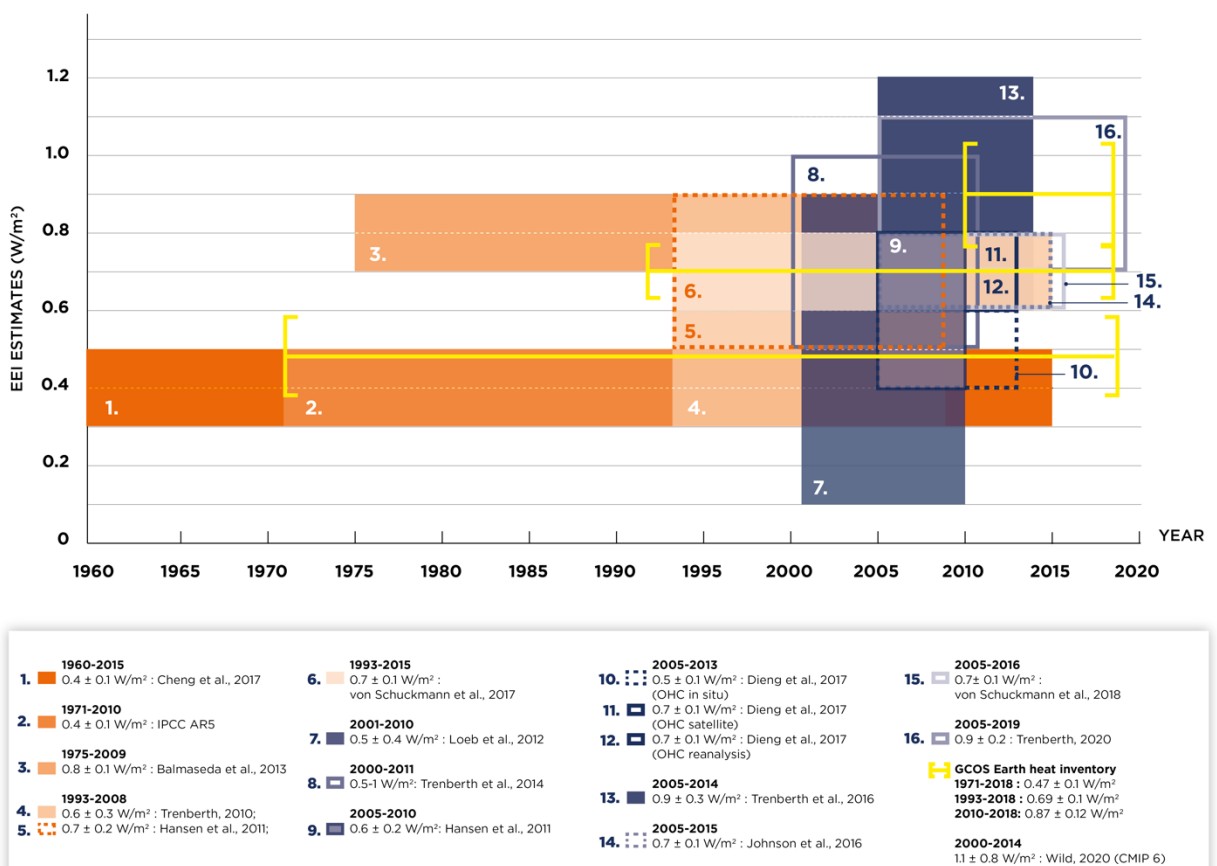


***Fig. 7:*** *Overview on EEI estimates as obtained from previous publications, and references are listed in the*
*figure legend. For IPCC AR5, Rhein et al. (2013) is used. The color bars take into account the uncertainty*
*ranges provided in each publication, respectively. For comparison, the estimates of our Earth heat*
*inventory based on the results of Fig. 6 have been added (yellow lines) for the periods 1971-2018, 1993-*
*2018 and 2010-2018, and the trends have been evaluated using a weighted least square fit (see von*
*Schuckmann and Le Traon, 2011 for details on the method).*

Rates of change derived from Fig. 6 are in agreement with previously published results for the
different periods (Fig. 7). Major disagreements occur for the estimate of Balmaseda et al. (2013)
which is obtained from an ocean reanalysis, and known to provide higher heat gain compared to
results derived strictly from observations (Meyssignac et al., 2019). Over the last quarter of a
decade this Earth heat inventory reports - in agreement with previous publications - an increased
rate of Earth heat uptake reaching up to $0.9$ W/m$^2$ (Fig. 7). This period is also characterized with
an increase in the availability and quality of the global climate observing system, particularly for
the past 2 decades. The heat inventory as obtained in this study reveals an EEI of $0.87 \pm 0.12$ W/m$^2$
over the period 2010-2018 - a period which experienced record levels of Earth surface warming,
and is ranked as the warmest decade relative to the reference period 1850-1900 (WMO, 2020).
Whether this increased rate can be attributed to an acceleration of global warming and Earth system
heat uptake (e.g. Cheng et al., 2018; WMO, 2020; Blunden and Arndt, 2019), or an induced
estimation bias due to the interplay between natural and anthropogenic driven variability (e.g.
Cazenave et al., 2014), or underestimated uncertainties in the historical record (e.g. Boyer et al.,
2016) needs further investigation.

The new multidisciplinary estimate obtained from a concerted international effort provides an
updated insight in where the heat is going from a positive EEI of $0.47 \pm 0.1$ W/m$^2$ for the period
1971-2018. Over the period 1971-2018 (2010-2018), 89% (90%) of the EEI is stored in the global
ocean, from which 52% (52%) is repartitioned in the upper 700m depth and 28% (30%) at
intermediate layers (700-2000m), and 9% (8%) in the deep ocean layer below 2000m depth.
Atmospheric warming amounts to 1% (2%) in the Earth heat inventory, the land heat gain with
6% (5%) and the heat uptake by the cryosphere with 4% (3%). These results show general
agreement with previous estimates (e.g. Rhein et al., 2013). Over the period 2010-2018, the EEI
amounts to $0.87 \pm 0.12$ W/m$^2$, indicating a rapid increase in EEI over the past decade. Note that a
near-global (60°N-60°S) area for the ocean heat uptake is used in this study, which could induce
a slight underestimation, and needs further evaluation in the future (see section 1). However, a test
using a single dataset (Cheng et al., 2017) indicates that the ocean contribution within 1960-2018
can increase by 1% if the full global ocean domain is used (not shown).

## 6. Conclusion

The UN 2030 Agenda for Sustainable Development states that climate change is "one of the greatest challenges of our time…" and warns "…the survival of many societies, and of the biological support systems of the planet, is at risk" (UNGA, 2015). The outcome document of the Rio+20 Conference, *The Future We Want*, defines climate change as "an inevitable and urgent global challenge with long-term implications for the sustainable development of all countries" (UNGA, 2012). The Paris agreement builds upon the United Nations Framework Convention on Climate Change (UN, 1992) and for the first time all nations agreed to undertake ambitious efforts to combat climate change, with the central aim to keep global temperature rise this century well below 2°C above pre-industrial levels and to limit the temperature increase even further to 1.5°C (UN, 2015). Article 14 of the Paris Agreement requires the *Conference of the Parties serving as the meeting of the Parties to the Paris Agreement* (CMA) to periodically take stock of the implementation of the Paris Agreement and to assess collective progress towards achieving the purpose of the Agreement and its long-term goals through the so called global stocktake based on best available science.

The EEI is the most critical number defining the prospects for continued global warming and climate change (Hansen et al., 2011, von Schuckmann et al., 2016), and we call for an implementation of the EEI into the global stocktake. The current positive EEI is understood to be foremost and primarily a result of increasing atmospheric greenhouse gases (IPCC, 2013), which have - according to the IPCC special report on Global Warming of 1.5 ºC - already 'caused approximately 1.0°C of global warming above pre-industrial levels, with a likely range of 0.8°C to 1.2°C' (IPCC, 2018). The IPCC special report further states with high confidence that 'global warming is likely to reach 1.5°C between 2030 and 2052 if it continues to increase at the current rate'. The EEI is the portion of the forcing that the Earth's climate system has not yet responded to (Hansen et al., 2005), and defines additional global warming that will occur without further change in forcing (Hansen et al., 2017). Our results show that EEI is not only continuing, it is increasing. Over the period 1971-2018 average EEI amounts to $0.47 \pm 0.12$ W/m$^2$, but amounts to $0.87 \pm 0.12$ W/m$^2$ during 2010-2018 (Fig. 8). Concurrently, acceleration of sea level rise (WCRP, 2018; Legelais et al., 2020), accelerated surface warming, record temperatures and sea ice loss in the Arctic (Richter-Menge et al., 2019; WMO, 2020; Blunden and Arndt, 2020) and ice loss from the Greenland ice sheet (King et al., 2020), and intensification of atmospheric warming near-surface and in the troposphere (Steiner et al., 2020) have been - for example - recently reported. To what degree these changes are intrinsically linked needs further evaluations.

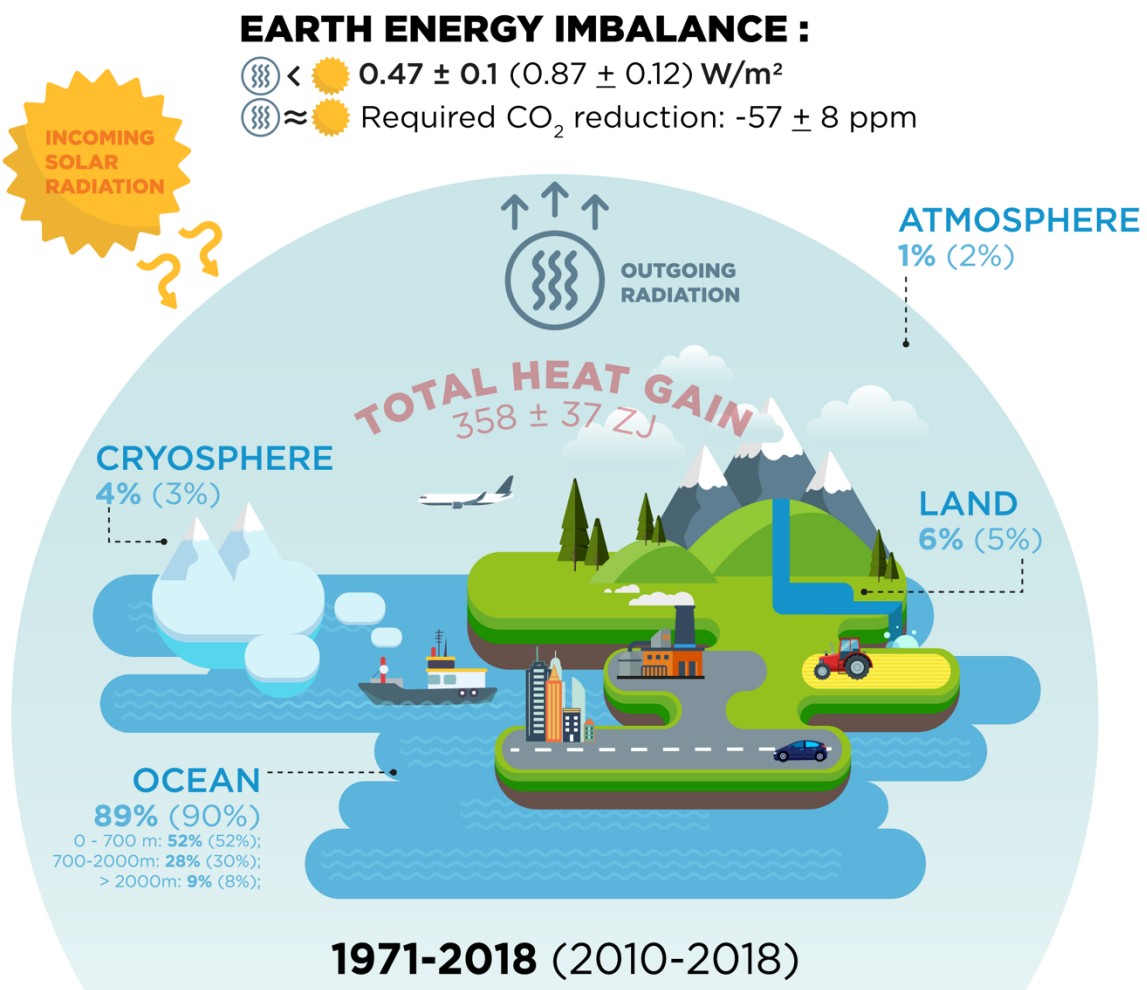

***Figure 8:*** *Schematic presentation on the Earth heat inventory for the current anthropogenic driven positive*
*Earth energy imbalance at the Top Of the Atmosphere (TOA). Relative partition (in %) of the Earth heat*
*inventory presented in Fig. 6 for the different components are given for the ocean (upper: 0-700m,*
*intermediate: 700-2000m, deep: > 2000m), land, cryosphere (grounded and floating ice) and atmosphere,*
*for the periods 1971-2018 and 2010-2018 (for the latter period values are provided in parentheses), as well*
*as for the EEI. The total heat gain (in red) over the period 1971-2018 is obtained from the Earth heat*
*inventory as presented in Fig. 6. To reduce the 2010-2018 EEI of $0.87 \pm 0.12$ W/m$^2$ to zero, current*
*atmospheric $CO_2$ would need to be reduced by $-57 \pm 8$ ppm (see text for more details).*

Global atmospheric CO2 concentration reached $407.38 \pm 0.1$ ppm averaged over 2018
(Friedlingstein et al., 2019) and $409.8 \pm 0.1$ ppm in 2019 (Blunden and Arndt, 2020). WMO (2020)
reports $CO_2$ concentrations at the Mauna Loa measurement platform of 411.75 ppm in February
2019, and 414.11 ppm in February 2020. Stabilization of climate, the goal of the universally agreed
UNFCCC (UN, 1992) and the Paris Agreement (UN, 2015), requires that EEI be reduced to
approximately zero to achieve Earth's system quasi-equilibrium. The change of heat radiation to
space for a given greenhouse gas change can be computed accurately. The amount of $CO_2$ in the
atmosphere would need to be reduced from 410 ppm to 353 ppm (i.e. a required reduction of -57
± 8 ppm) to increase heat radiation to space by 0.87 W/m$^2$, bringing Earth back towards energy
balance (Fig. 8), where we have used the analytic formulae of Hansen et al. (2000) for this
estimation. Atmospheric $CO_2$ was last 350 ppm in the year 1988, and global Earth's surface
temperature was then +0.5°C relative to the pre-industrial period (relative to the 1880-1920 mean)
(Hansen et al., 2017; Friedlingstein et al., 2019). In principle, we could reduce other greenhouse
gases, and thus require a less stringent reduction of $CO_2$. However, as discussed by Hansen et al.
(2017), some continuing increase of $N_2O$, whose emissions are associated with food production,
seems inevitable, so there is little prospect for much net reduction of non-$CO_2$ greenhouse gases,
and thus, the main burden for climate stabilization falls on $CO_2$ reduction. This simple number,
EEI, is the most fundamental metric that the scientific community and public must be aware of, as
the measure of how well the world is doing in the task of bringing climate change under control
(Fig. 8).

This community effort also addresses gaps for the evolution of future observing systems for a
robust and continued assessment of the Earth heat inventory, and its different components.
Immediate priorities include the maintenance and extension of the global climate observing system
to assure a continuous monitoring of the Earth heat inventory, and to reduce the uncertainties. For
the global ocean observing system, the core Argo sampling needs to be sustained, and
complemented by remote sensing data. Extensions such as into the deep ocean layer need to be
further fostered (Desbruyères et al., 2017; Johnson et al., 2015), and technical developments for
the measurements under ice and in shallower areas need to be sustained and extended. Moreover,
continued efforts are needed to further advance bias correction methodologies, uncertainty
evaluations and data processing of the historical dataset.

In order to allow for improvements on the present estimates of changes in the continental heat and
to ensure that the database is continued into the future, an international, coordinated effort is
needed to increase the number of subsurface temperature data from BTPs at additional locations
around the world, in particular in the southern hemisphere. Additionally, repeated monitoring
(after a few decades) of existing boreholes should help reduce uncertainties at individual sites.
Such data should be shared through an open platform.

For the atmosphere, the continuation of operational satellite- and ground-based observations is
important but the foremost need is sustaining and enhancing a coherent long-term monitoring
system for the provision of climate data records of essential climate variables. GNSS radio
occultation observations and reference radiosonde stations within the Global Climate Observing
System (GCOS) Reference Upper Air Network (GRUAN) are regarded as climate benchmark
observations. Operational radio occultation missions for continuous global climate observations
need to be maintained and expanded, ensuring global coverage over all local times, as the central
node of a global climate observing system.

For the cryosphere, sustained remote sensing for all of the cryosphere components is key to
quantifying future changes over these vast and inaccessible regions, but must be complemented by
in-situ observations for calibration and validation. For sea-ice, the albedo, the area and ice
thickness are all essential, with ice-thickness being particularly challenging to quantify with
remote sensing alone. For ice sheets and glaciers, reliable gravimetric measurements, ice thickness
and extent, snow/firn thickness and density are essential to quantify changes in mass balance of
grounded and floating ice. We highlight Antarctic sea-ice change and warming of firn as terms
that are poorly constrained or have not significantly contributed to this assessment but may become
important over the coming decades. Similarly, there exists the possibility for rapid change
associated with positive ice dynamical feedbacks at the marine margins of the Greenland and
Antarctic ice sheets. Sustained monitoring of each of these components will, therefore, serve the
dual purpose of furthering understanding of the dynamics and quantifying the contribution to
Earth's energy budget. In addition to data collection, open access to the data and data synthesis
products as well as coordinated international efforts are key to the continued monitoring of the ice
loss from the cryosphere and related energy uptake.

Sustained and improved observations to quantify Earth's changing energy inventory are also
critical to the development of improved physical models of the climate system, including both data
assimilation efforts that help us to understand past changes and predictions (Storto et al., 2019)
and climate models used to provide projections of future climate change (Eyring et al., 2019). For
example, atmospheric reanalyses have shown to be a valuable tool for investigating past changes
in the EEI (Allan et al., 2014) and ocean reanalyses have proven useful in estimating rates of ocean
heating on annual and sub-annual timescales by reducing observational noise (Trenberth et al.,
2016). Furthermore, both reanalyses and climate models can provide information to assess current
observing capabilities (Fujii et al., 2019) and improve uncertainty estimates in the different
components of Earth's energy inventory (Allison et al., 2019). Future priorities for expanding the
observing system to improve future estimates of EEI should be cognizant of the expected evolution
of the climate change signal, drawing on evidence from observations, models and theory
(Meyssignac et al., 2019; Palmer et al., 2019).

A continuous effort to regularly update the Earth heat inventory is important to quantify how much
and where heat accumulated from climate change is stored in the climate system. The Earth heat
inventory crosses multi-disciplinary boundaries, and calls for the inclusion of new science
knowledge from the different disciplines involved, including the evolution of climate observing
systems and associated data products, uncertainty evaluations and processing tools. The results
provide indications that a redistribution and conversion of energy in the form of heat is taking
place in the different components of the Earth system, particularly within the ocean, and that EEI
has increased over the past decade. The outcomes have further demonstrated how we are able to
evolve our estimates for the Earth heat inventory while bringing together different expertise and

major climate science advancements through a concerted international effort. All of these component estimates are at the leading edge of climate science. Their union has provided a new and unique insight on the inventory of heat in the Earth system, its evolution over time, as well as a revision of the absolute values. The data product of this effort is made available and can be thus used for model validation purposes.

This study has demonstrated the unique value of such a concerted international effort, and we thus call for a regular evaluation of the Earth heat inventory. This first attempt presented here has been focused on the global area average only, and evolving into regional heat storage and redistribution, the inclusion of various time-scales (e.g. seasonal, year-to-year), and other climate study tools (e.g. indirect methods, ocean reanalyses) would be an important asset of this much needed regular international framework for the Earth heat inventory. This would also respond directly to the request of GCOS to establish the observational requirements needed to monitor the Earth's cycles and the global energy budget. The outcome of this study will therefore directly feed into GCOS' assessment of the status of the global climate observing system due in 2021 which is the basis for the next implementation plan. These identified observation requirements will guide the development of the next generation of in-situ and satellite global climate observations by all national meteorological services and space agencies and other oceanic and terrestrial networks.

**Data availability:** The time series of the Earth heat inventory are published at DKRZ (https://www.dkrz.de/) under the doi: https://doi.org/10.26050/WDCC/GCOS_EHI_EXP_v2 (von Schuckmann et al., 2020). The data contain an updated international assessment of ocean warming estimates, and new and updated estimates of heat gain in the atmosphere, cryosphere and land over the period 1960-2018. This published dataset has been used to build the basis for Figure 6 of this manuscript. The ocean warming estimate is based on an international assessment of 15 different in situ data-based ocean products as presented in section 1. The new estimate of the atmospheric heat content is fully described in section 2, and is backboned on a combined use of atmospheric reanalyses, multi-satellite data and radiosonde records, and microwave sounding techniques. The land heat storage time series as presented in section 3 relies on borehole data. The heat available to account for cryosphere loss is presented in section 4, and is based on a combined use of model results and observations to obtain estimates of major cryosphere components such as polar ice sheets, Arctic sea-ice and glaciers.

**GCOS Earth heat inventory team: Author contributions**

Coordination: Karina von Schuckmann[1], Lijing Cheng[2,28], Matthew D. Palmer[3], Caterina Tassone[5], Valentin Aich[5]

Ocean: Karina von Schuckmann[1], Lijing Cheng[2,28], Tim Boyer[8], Damien Desbruyères[9], Catia Domingues[10,11], John Gilson[13], Masayoshi Ishii[16], Gregory C. Johnson[17], Rachel E. Killick[3], Brian A. King[10], Nicolas Kolodziejczyk[18], John Lyman[17], Maeva Monier[20], Didier Paolo Monselesan[21], Sarah Purkey[6], Dean Roemmich[6], Susan E. Wijffels[21,26]

Atmosphere: Gottfried Kirchengast[14], Maximilian Gorfer[14], Andrea K. Steiner[14], Michael Mayer[15,29], Leopold Haimberger[15]

Land: Almudena García-García[7], Francisco José Cuesta-Valero[7,27], Hugo Beltrami[7], Sonia I. Seneviratne[23], Pierre Gentine[12]

Cryosphere: Fiammetta Straneo[6], Susheel Adusumilli[6], Donald A. Slater[6], Mary-Louise Timmermans[25], Ben Marzeion[19], Axel Schweiger[22], Andrew Shepherd[24]

Earth energy inventory: all authors, with specific contributions from Karina von Schuckmann[1], Almudena García-García[7], Francisco José Cuesta-Valero[7,27], James Hansen[4], Maeva Monier[19]

Conclusion: all authors, with specific contributions from Karina von Schuckmann[1], James Hansen[4], Caterina Tassone[5], Valentin Aich[5], Lijing Cheng[2,28], Matthew D. Palmer[3], Gottfried Kirchengast[14], Andrea K. Steiner[14], Almudena García-García[7], Francisco José Cuesta-Valero[7,27], Hugo Beltrami[7], Fiammetta Straneo[6]

[1]Mercator Ocean International, France
[2]Institute of Atmospheric Physics, Chinese Academy of Sciences, China
[3]Met Office Hadley Centre, UK
[4]Columbia University Earth Institute, USA
[5]WMO/GCOS, Switzerland
[6]Scripps Institution of Oceanography, UCSD, San Diego, CA, USA
[7]Climate & Atmospheric Sciences Institute, St. Francis Xavier University, NS, Canada
[8]NOAA's National Centers for Environmental Information
[9]Ifremer, University of Brest, CNRS, IRD, Laboratoire d'Océanographie Physique et Spatiale, France
[10]National Oceanographic Centre, UK
[11]ARC Centre of Excellence for Climate Extremes, University of Tasmania, Hobart, Tasmania, Australia
[12]Earth and Environmental Engineering in the School of Engineering and Applied Sciences, Columbia University, USA
[13]University of California, USA
[14]Wegener Center for Climate and Global Change and Institute of Physics, University of Graz, Austria

[15]Department of Meteorology and Geophysics, University of Vienna, Austria

[16]Department of Atmosphere, Ocean and Earth System Modeling Research, Meteorological Research Institute, Japan

[17]NOAA, Pacific Marine Environmental Laboratory, USA

[18]University of Brest, CNRS, IRD, Ifremer, Laboratoire d'Océanographie Physique et Spatiale, IUEM, France

[19]Institute of Geography and MARUM-Center for Marine Environmental Sciences, University of Bremen, Germany

[20]CELAD/Mercator Ocean International, France

[21]CSIRO Oceans and Atmosphere, Hobart, Tasmania, Australia

[22]Polar Science Center, Applied Physics Laboratory, University of Washington, Seattle, USA

[23]Institute for Atmospheric and Climate Science, ETH, Switzerland

[24]Center for Polar Observation and Modeling, University of Leeds, UK

[25]Department of Earth and Planetary Sciences, Yale University, New Haven, USA

[26]Woods Hole Oceanographic Institution, Massachusetts, United States

[27]Environmental Sciences Program, Memorial University of Newfoundland, NL, Canada

[28]Center for Ocean Mega-Science, Chinese Academy of Sciences, Qingdao, China, 266071

[29]European Centre for Medium-Range Weather Forecasts, Reading, UK

## **Acknowledgements:**

This research benefited from long-term attention by GCOS and builds on initial work carried out under the WCRP core projects CLIVAR, GEWEX, CliC and SPARC.

MDP and REK were supported by the Met Office Hadley Centre Climate Programme funded by the BEIS and Defra.

Ocean: PMEL contribution number 5053; Funding: CMD was supported by an ARC Future Fellowship (FT130101532). L.Cheng is supported by Key Deployment Project of Centre for Ocean Mega-Research of Science, CAS (COMS2019Q01).

Atmosphere: The authors express their gratitude to SPARC for supporting this activity under sponsorship of the WCRP. M. Gorfer was supported by WEGC atmospheric remote sensing and climate system research group young scientist funds. M. Mayer was supported by Austrian Science Fund project P33177. We acknowledge the WEGC EOPAC team for providing the OPSv5.6 RO data (available online at https://doi.org/10.25364/WEGC/OPS5.6:2019.1) as well as quality-processed Vaisala RS data, UCAR/CDAAC (Boulder, CO, USA) for access to RO phase and orbit data, RSS (Santa Rosa, CA, USA) for providing MSU V4.0 data, ECMWF (Reading, UK) for access to operational analysis and forecast data, ERA5 reanalysis data, and RS data from the ERA-Interim archive, JMA (Tokyo, Japan) for provision of the JRA55 and JRA55C reanalysis data, and NASA GMAO (Greenbelt, MD, USA) for access of the MERRA-2 reanalysis data.

Land: This work was supported by grants from the National Sciences and Engineering Research Council of Canada Discovery Grant (NSERC DG 140576948) and the Canada Research Chairs Program (CRC 230687) to H. Beltrami. Almudena García-García and Francisco José Cuesta-Valero are funded by Beltrami's CRC program, the School of Graduate Studies at Memorial University of Newfoundland and the Research Office at St. Francis Xavier University

Ice: This work was supported by CliC (Climate and Cryosphere Project of the World Climate Research
Program). FS acknowledges support by NSF OCE 1657601. SA was supported by the NASA grant
80NSSC18K1424.

We also would like to thank Marianne Nail (https://girlsmakesense.com/) for her support to the graphical
development of Fig. 7 and 8.

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
