# Peer review of "Heat stored in the Earth system: Where does the energy go?"

_Earth System Science Data, 2019_

## Referee Comment (RC1) · Anonymous Referee #1 · 24 Apr 2020

Review of: Heat stored in the Earth system: Where does the energy go?

General Comments:

The paper provides a very nice update on the Earth's heat inventory for 1960-2018. It is a collaborative effort involving many authors who are experts in the various Earth subsystems in which heat storage occurs. One of the main findings from this analysis is a decrease in the contribution of ocean to the Earth heat inventory (89%) compared to prior assessments (93%), and a doubling of the land contribution (6% vs 3%). The latter is based upon recent analyses of data from FluxNet, geothermal data and model simulations. The paper also finds that the ocean heating rate has doubled since the

beginning of the altimeter era (1993-2018) compared to the "historical" period (1960-2018), and that the contribution of ocean heating in intermediate (700-2000 m) and deep (> 2000 m) ocean layers is notably greater for 2000-2018 compared to 1960-2018.

These new results are interesting and worthy of publication. However, it is unclear why the paper does not describe or discuss in any detail the geographical distribution of heat storage in the ocean. The global vertical distribution is discussed extensively but wouldn't the geographical distribution also be worth mentioning, particularly given the paper's title? Admittedly, this may only be feasible during the Argo period because of its better geographical sampling compared to earlier periods, but it seems appropriate to include a short discussion about this nonetheless.

Recommendation: Accept with some minor revisions.

Specific Comments:

Line 78: Hansen et al., 2005 is not in the reference list.

Line 151-153: Sentence beginning with "However". Please provide a reference or two supporting this statement.

Table 1 (second-to-last row): "0.7-0.9 +/- 0.1 Wm-2. The range is greater than the uncertainty. This implies the uncertainty is too small.

Figure 1: The figure would be clearer if colors were more distinct. Consider using more than just different shades of blue.

Lines 281-283: The trends given in the caption should appear in a separate table.

Figure 2: Are the trends for the ocean area only or are they averages over the entire surface area of the globe (as they presumably are in Table 1)? A common reference throughout the paper would be helpful in order to compare the magnitudes of heat storage in different parts of the Earth system.

Lines 392-393: cv is defined twice.

Lines 449-451: Is it necessary to mention everyone?

Lines 498-502: This is a very long sentence. Consider breaking it up into two or more sentences.

Lines 502-504 "These radiatively relevant processes include the stability and extent of the continental areas occupied by permafrost soils."

Awkward sentence. Consider rewording. For example, "the stability and extent of the continental areas occupied by permafrost soils" are not "processes".

Line 555: No "," after "Such records".

Line 556: What does "beyond the observational record" refer to? Perhaps you mean "prior to the observational record"? Lines 687-688: Do you mean 40 years instead of 40 decades?

Figure 7: The color for the atmosphere contribution is inconsistent between the pie chart (green) and label (blue).

[Figure]

---

## Referee Comment (RC2) · Anonymous Referee #2 · 26 Apr 2020

Review ESSD-2019-255, Energy budget

Overall a good important energy inventory. Unique coordinated comprehensive approach across ocean, atmosphere, land and ice. Valuable product in its own right, but also as a review of each component. A few small suggestions and changes below, along with a call for a clearer vision of future efforts.

Lines 53, 54: Good list but with redundancy? "warming oceans, atmosphere and land," = "rising temperatures". Need clarification.

Line 55, 56: Reader does not need to see 'international' twice in the same sentence.

Line 64: "There is a regime shift" Write instead 'there was a regime shift'. Given data up to 2018, authors would do well to adopt past tense throughout? This concept of a regime shift gets lost later in the document?

Lines 92, 93: "represents a more robust measure of the rate of climate change that is more indicative of the time-evolution" more robust but also more indicative? Reader does not need both 'more'.

Line 140: capitalize Arctic.

Lines 181-185: Too much redundancy in this list of ocean impact.

Line 283: "values are given for the ocean surface." Units of $J/m^2$, so column-integrated numbers. Understand 0-300, 0-700 and 0-2000, but how do the authors extrapolate 700-2000 to surface heat flux values? 0-2000 minus 0-700?

Lines 322, 323: "heat sequestration into the deeper ocean layers took place over the past 6 decades" Because all OHC trend values in Figure 2 are positive? Do readers need to see units on ordinate of Figure 2, e.g as $W / m^2$?

Lines 384, 385: "Fsnow cools the high latitude ocean" ? If Fsnow represents an upward heat flux that warms high latitude atmosphere during snow formation, how does it also cool high latitude ocean? Snow falling on the ocean requires heat from the ocean to melt, but does that energy to melt = Fsnow? Some careful changes here could allay confusion.

Line 445: Confusion about "third … dataset" here because in prior paragraph we just read about three observation techniques (ERA, JRA and Merra)? Soon after, in legend for Figure 3, reader encounters "four different reanalyses and two (left) or three (right, plus MSU) different observational datasets" Need some careful enumeration ….

Lines 491-497: Good summary but redundant with introductory text? Remnant from a prior stand-alone land product? Not needed here?

Line 509: "extreme heat events" heat and drought?

Line 585: Not sure why uncertainty (95% CI) should increase with time? These represent cumulative uncertainties?

Line 637: "snow and permafrost". But, energetics of permafrost thaw already included in land heat estimates?

Line 687, 688: "past 40 decades" I hope the authors mean the past 4 decades?

Line 743: Useful important figure but it would help visual memory of this reader if colors in the side-by-side pie chart (Figure 7) matched those used in the time series chart (Figure 6).

Line 749: Conceptual discontinuity with prior section 5; here the text jumps to what next. Make this a separate section? Perhaps summarize major factors that contribute to overall uncertainty (well itemized in each individual section but not yet presented from an overall e.g. GCOS viewpoint)? Then move to recommendations? Lines 749 to 773 read like the standard GCOS wish list. No priorities? Nothing most urgent, e.g. to resolve/reduce key uncertainties?

Lines 772, 773: "remote sensing measurements have to be calibrated and validated by in situ measurements." By now basically a platitude? Haven't we been saying / writing this recommendation for decades? What new from this work heightens or refines that recommendation? I do not expect a consensus to arise from the full list of contributors, but here the lead author(s) could inject expert opinions?

Lines 775 to 783: Readers might expect a stronger outcome / summary? Builds on large communities for each component, needs more or higher level of coordination? What made it different from (improved over) prior efforts? Here, EEI calculated through 2018 (with some extrapolations for some components since 2016). How fast does the system evolve? How fast do our observations evolve? Next inventory in 5 years? Two years? What does the science require, compared to what the observations can provide? Vast amount of work summarized here but the conclusion leaves readers wondering what next?

Numerous small typos that I hope the proofreaders will correct.

---

## Short Comment (SC1) · 21 May 2020

Summary

The manuscript provides a WRCP-SPARC group summary of the status of our understanding of Earth's Energy Imbalance since 1960 and covers much of the recent progress in documenting the component level evolution of energy uptake, including ocean, atmosphere, cryosphere, and land. An effort is made to quantify some of the

uncertainties. I find the manuscript to be reasonably well organized and clear in scope. I do have numerous suggestions to improve the writing, which is in instances awkward, but these are generally minor. I also strongly suggest that the tables be reconsidered in favor of a graphical summary such as the one below, which I believe will be much more effective at conveying the key points. I also think additional context should be added, particularly to the section on boreholes, to address the limitations of this data source that I believe go largely unmentioned. I have similar concerns regarding the ocean heat content section. Lastly I also feel some effort should be made to minimize the aliasing of internal variability onto the trends.

Major Issues

1) The tables are largely ineffectual at conveying the key points. See suggestions below.

2) Basic characteristics of the borehole data go unmentioned and associated caveats in forming an estimate of global-land heat uptake go unaddressed.

3) The role of sampling uncertainty in the ocean heat content estimates gets swept under the rug when the multi-estimate average is performed and Figure 1/2 and their discussion gives an overly confident depiction in my view. This issue should be raised and emphasized in my opinion - as it is a major caveat to the summaries.

4) No effort is made to account for internal variability in accounting for trends. This is particularly relevant for the atmosphere where the role of ENSO is strong. I suggest some effort be made and a number of reasonable approaches exist.

Minor Issues

78: "internal climate modes can temporarily alter the energy balance for periods of sub-monthly to several decades"; deserves several references

92-93: this statement also deserves several references - e.g. Cheng et al EOS

101: there is substantial noise in this equation and this should be acknowledged with an additional term, e.g. epsilon, even in a conceptual model since it is unavoidable.

132: Table 1 would be more useful as a figure such as in the ocean heat uptake summary below as it provides a visual for both period and magnitude/uncertainty:

ï£ijhttp://www.realclimate.org/images//oh_uptake2.png

183: I find "change in behavior" to be vague and unclear.

187: I find "Implications of ocean warming. . ., and have in turn impacted the ocean itself" to be both unclear and poorly related to the following sentence where the impacts on the ocean go unmentioned and thus are not examples.

194: "challenges of Ocean Heat Content (OHC) estimates" is poor grammar. what I think is mean is "opportunities and challenges in forming OHC estimates. . .".

196. Remove "Early" since before 1900 is already stated.

215: "further challenged the global scale ocean heat content estimate" perhaps should read "posed a further challenge for the global scale estimation of ocean heat content"

216: "correct biases"

222: "with satellite-derived. . ."

233: "The opportunity for improved OHC estimation provided by Argo is tremendous"

239: "allow further observing system recommendations" could read "inform further observing system recommendations"

258: 2018).

265: What initiative?

268: We do (present tense?)

271: Is an "ensemble mean and standard deviation of near-global " an improvement?

The assumption is that all errors are random and not systematic. But are they? I think the case would need to be made, and I don't think it is largely true.

273: "is needed to include all missing in situ-based products" is not clear what is being referred to? what are the main data sources that have yet to be included (in reanalyses for example)? Figure 1: What exactly is in the sigma? XBT drop rate uncertainties? sampling uncertainties? I sense that the answer is "none" but rather the spread in the derivations of previous products, which themselves have much larger uncertainties and that averaging together does not reduce. This is a point worth emphasizing as Fig 1 seem overconfident as-is. I thing that a true "uncertainty" bound be shown somehow (whiskers?). And I suggest the reference to "values are given for the ocean surface" be made clearer as the figure itself obviously doesn't deal with the ocean surface (just the area used for the heating rates). Figure 2: As with Figure 1 there is an underlying sampling/data uncertainty that is not dealt with but is very relevant to the values plotted and should be shown. Ultimately I think this uncertainty creates an important caveat for the results. Also the figure/caption lack units).

311: But this result is already well known?

370: Eq (1) seems off as the surface enthalpy fluxes for P and E are already contained in FS, unless what is meant is the very small term of the sensible heat of rainfall? (other components of enthalpy such as evaporation are already in FS). Given the description later (in lines 388-90) I think sensible heat is the correct term? as enthalpy also entails the latent component.

381: Why "global" evaporation and precipitation?

398: "account for"

406: "neglect"

425: "used" . . .

Figure 3: The caption does not adequately distinguish between the terms plotted in

the middle and bottom row. Perhaps it could better emphasize that one is the anomaly while the other is the annual tendency (change and trends can be taken as synonymous).

476: An important point is raised here. Clearly El Niño is aliasing on to the trends to some extent. Shouldn't an effort be made to remove such effects? Various approaches have been tried in the literature and I'm not advocating for any single one, but to make no effort seems lazy.

544: There is an odd symbol after "condition."?

555: The entire discussion is very generic here. What is the spatial sampling of boreholes that can be used and what temporal resolution for GSTHs can they provide and at what spatial sampling / accuracy can they be used. How does groundwater corrupt/influence inferred histories? What validation of the reconstructions exist and what does all of this reveal that is both new and robust about "the evolution of the climate system".

565. How is a global estimate inferred from what must be very discrete observations?

Table 2: Again I think Table 2 would be best as a figure such as suggested for Table 1.

639: Remove "in sea-ice extent" as it is redundant.

624: "with ice loss"

652: "which are based on a modeled surface mass balance combined with..."

654: "estimate of floating ..."

740: "needs further evaluation"

761: There is oddly no mention of the need to develop modeling/assimilation capabilities as a priority. The authors may view these as distinct from the observational imperatives however in my view they are not, as they are central to the generation

of ocean/atmosphere/cryosphere models that are vital for improving the reanalyses used. The same can be argued for terrestrial system models given the poor coverage of borehole measurements.

[Figure]

**Ocean heat uptake estimates over time**
**(averaged over whole surface)**

[Figure]

Fig. 1.

---

## Author Comment (AC1) · 22 Jul 2020

Heat stored in the Earth system: Where does the energy go? The GCOS Earth heat inventory team Point-by-point reply to review

Anonymous Referee #1

Review of: Heat stored in the Earth system: Where does the energy go? General Comments: The paper provides a very nice update on the Earth's heat inventory for 1960-2018. It is a collaborative effort involving many authors who are experts in the various Earth subsystems in which heat storage occurs. One of the main findings from

this analysis is a decrease in the contribution of ocean to the Earth heat inventory (89%) compared to prior assessments (93%), and a doubling of the land contribution (6% vs 3%). The latter is based upon recent analyses of data from FluxNet, geothermal data and model simulations. The paper also finds that the ocean heating rate has doubled since the beginning of the altimeter era (1993-2018) compared to the "historical" period (1960- 2018), and that the contribution of ocean heating in intermediate (700-2000 m) and deep (> 2000 m) ocean layers is notably greater for 2000-2018 compared to 1960-2018. These new results are interesting and worthy of publication. However, it is unclear why the paper does not describe or discuss in any detail the geographical distribution of heat storage in the ocean. The global vertical distribution is discussed extensively but wouldn't the geographical distribution also be worth mentioning, particularly given the paper's title? Admittedly, this may only be feasible during the Argo period because of its better geographical sampling compared to earlier periods, but it seems appropriate to include a short discussion about this nonetheless.

We thank the reviewer for his comment. While discussing the establishment of this community work, the group under GCOS has extensively discussed the content of this work, including a discussion on regional distributions, other time-scales (e.g. seasonal cycle, year-to-year variations, etc. . .). There had been overall agreement that this first publication should be limited to the area averaged (global) estimate only, with focus on the long-term change. We hope that a continuation of this community activity will be possible in the future, including extensions of the discussions, such as the regional distribution - which should be then not limited to the assessment of the regional ocean warming signal only. We will thus not include a specific discussion in this manuscript on regional ocean heat content distribution.

Recommendation: Accept with some minor revisions.

Thank you very much, and we provide below a point-by-point reply to the specific issues raised by the reviewer.

Specific Comments:

Line 78: Hansen et al., 2005 is not in the reference list.

Thank you for spotting this, and we have now added this reference into the list.

Line 151-153: Sentence beginning with "However". Please provide a reference or two supporting this statement. We have added the reference Hansen et al., 2011.

Table 1 (second-to-last row): "0.7-0.9 +/- 0.1 Wm-2. The range is greater than the uncertainty. This implies the uncertainty is too small. We have used the uncertainty range as provided in the cited references, and providing more uncertainty evaluations from these specific studies is out of the scope of this paper.

Figure 1: The figure would be clearer if colors were more distinct. Consider using more than just different shades of blue. We have changed the colors.

Lines 281-283: The trends given in the caption should appear in a separate table. We have added the values now in the figure legend.

Figure 2: Are the trends for the ocean area only or are they averages over the entire surface area of the globe (as they presumably are in Table 1)? A common reference throughout the paper would be helpful in order to compare the magnitudes of heat storage in different parts of the Earth system. There is a clarification of the area used in the figure caption: 'Note that values are given for the ocean surface.'

Lines 392-393: cv is defined twice. Thank you, we have removed the repetition.

Lines 449-451: Is it necessary to mention everyone? No, thank you for spotting this error arised from the use of a reference management system.

Lines 498-502: This is a very long sentence. Consider breaking it up into two or more sentences. We have modified this sentence in the new version of the manuscript.

Lines 502-504 "These radiatively relevant processes include the stability and extent

of the continental areas occupied by permafrost soils." Awkward sentence. Consider rewording. For example, "the stability and extent of the continental areas occupied by permafrost soils" are not "processes". We have modified this sentence according to the reviewer's comment.

Line 555: No "," after "Such records". We have corrected this typo.

Line 556: What does "beyond the observational record" refer to? Perhaps you mean "prior to the observational record"? Indeed, we meant before the observational period. We have changed that on the text accordingly.

Lines 687-688: Do you mean 40 years instead of 40 decades? Yes, thank you, we have modified accordingly (4 decades).

Figure 7: The color for the atmosphere contribution is inconsistent between the pie chart (green) and label (blue). We have re-drawn Fig. 7 in order to provide coherence with Fig. 6 for the color choice, and the problem the reviewer has raised is now fixed. Thank you.

%%%%%%%%%%%%%%%%%%%%%%%%%%%%%%%%%%%%%%%%%%%%%%

Anonymous Referee #2

Overall a good important energy inventory. Unique coordinated comprehensive approach across ocean, atmosphere, land and ice. Valuable product in its own right, but also as a review of each component. A few small suggestions and changes attached, along with a call for a clearer vision of future efforts. Detailed comments in supplement. Please also note the supplement to this comment: https://www.earth-syst-sci-data-discuss.net/essd-2019-255/essd-2019-255-RC2- supplement.pdf We thank the reviewer, which helps to improve the manuscript, and we provide a point-by-point reply to all comments raised by this reviewer.

SUPPLEMENT:

Review ESSD-2019-255, Energy budget Overall a good important energy inventory. Unique coordinated comprehensive approach across ocean, atmosphere, land and ice. Valuable product in its own right, but also as a review of each component. A few small suggestions and changes below, along with a call for a clearer vision of future efforts.

Lines 53, 54: Good list but with redundancy? "warming oceans, atmosphere and land," = "rising temperatures". Need clarification. We agree, and we have thus modified, i.e. use 'rising surface temperature' instead to be more specific and to avoid redundancy.

Line 55, 56: Reader does not need to see 'international' twice in the same sentence. Thank you, and corrected (removed the second 'international').

Line 64: "There is a regime shift" Write instead 'there was a regime shift'. Given data up to 2018, authors would do well to adopt past tense throughout? This concept of a regime shift gets lost later in the document? We have modified 'regime shift' into 'change'. We remain with past tense throughout the document.

Lines 92, 93: "represents a more robust measure of the rate of climate change that is more indicative of the time-evolution" more robust but also more indicative? Reader does not need both 'more'. Thank you and we have removed the first 'more'.

Line 140: capitalize Arctic. Done.

Lines 181-185: Too much redundancy in this list of ocean impact. We went back to the list, but we cannot spot the redundancy the reviewer is referring to. A sequence as 'Together with ocean acidification and deoxygenation, ocean warming can lead to' might give the impression for redundancy, but this is listed here again because impacts on ecosystems is only seen in the line of multiple lines of impact from ocean warming, de-oxygenation and ocean acidification, and for this purpose we do not want to remove to assure correctness.

Line 283: "values are given for the ocean surface." Units of J/m2, so column-integrated

numbers. Understand 0-300, 0-700 and 0-2000, but how do the authors extrapolate 700-2000 to surface heat flux values? 0-2000 minus 0-700? There is a misunderstanding, and this addition is given in terms of using / transferring into W/m2 while using not the entire Earth's surface, but the ocean surface only (which impacts by a factor of 0.7). We prefer to keep this addition, also in the light of comments from reviewer 1.

Lines 322, 323: "heat sequestration into the deeper ocean layers took place over the past 6 decades" Because all OHC trend values in Figure 2 are positive? Do readers need to see units on ordinate of Figure 2, e.g as W / m2? To better clarify, we have modified the sentence accordingly: Moreover, there is a clear indication that heat sequestration into the deeper ocean layers below 700m depth took place over the past 6 decades linked to an increase of OHC trends over time (Fig. 2).

Lines 384, 385: "Fsnow cools the high latitude ocean" ? If Fsnow represents an upward heat flux that warms high latitude atmosphere during snow formation, how does it also cool high latitude ocean? Snow falling on the ocean requires heat from the ocean to melt, but does that energy to melt = Fsnow? Some careful changes here could allay confusion. Yes, the heat it takes to melt snow falling into the ocean is Fsnow. This is also the amount of additional latent heat release in the atmosphere that occurs through freezing. We modified the concerning paragraph to clarify this.

Line 445: Confusion about "third ... dataset" here because in prior paragraph we just read about three observation techniques (ERA, JRA and Merra)? Soon after, in legend for Figure 3, reader encounters "four different reanalyses and two (left) or three (right, plus MSU) different observational datasets" Need some careful enumeration .... We thank the reviewer for this comment, and this is clarified now.

Lines 491-497: Good summary but redundant with introductory text? Remnant from a prior stand-alone land product? Not needed here? The reviewer is right, we have removed this small introduction from the text.

Line 509: "extreme heat events" heat and drought? We meant extreme temperature events. We have clarified this in the text.

Line 585: Not sure why uncertainty (95% CI) should increase with time? These represent cumulative uncertainties? The reviewer refers to the caption on Figure 5. In this figure, we represent the cumulative heat in the continental subsurface based on the global surface heat flux estimates displayed in Figure 4, which assumes zero heat storage in 1960. That is, the 95% CI in Figure 5 represents the cumulative heat uncertainty estimated from the 95% CI for the continental heat flux shown in Figure 4.

Line 637: "snow and permafrost". But, energetics of permafrost thaw already included in land heat estimates? Land heat estimates account for the warming of the land but not for the latent heat uptake by the thawing permafrost. This is now clarified within the cryosphere section text on permafrost.

Line 687, 688: "past 40 decades" I hope the authors mean the past 4 decades? Yes, thank you, and also raised by reviewer 1, we have fixed this now.

Line 743: Useful important figure but it would help visual memory of this reader if colors in the side-by-side pie chart (Figure 7) matched those used in the time series chart (Figure 6). We agree with the reviewers comment, and have modified fig. 7 accordingly.

Line 749: Conceptual discontinuity with prior section 5; here the text jumps to what next. Make this a separate section? Perhaps summarize major factors that contribute to overall uncertainty (well itemized in each individual section but not yet presented from an overall e.g. GCOS viewpoint)? Then move to recommendations? We agree, and we have considerably modified the final section, and introduced a new section 6 Conclusion

Lines 749 to 773 read like the standard GCOS wish list. No priorities? Nothing most urgent, e.g. to resolve/reduce key uncertainties? According to the reply to the above

comment, we have added a new paragraph into the conclusion, which we hope will also reply to this comment.

Lines 772, 773: "remote sensing measurements have to be calibrated and validated by in situ measurements." By now basically a platitude? Haven't we been saying / writing this recommendation for decades? What new from this work heightens or refines that recommendation? I do not expect a consensus to arise from the full list of contributors, but here the lead author(s) could inject expert opinions? We would like to remain with this statement in the text, even if it is a repetition over years - but given various discussions, we think that there is a need to continue to highlight the importance of in situ observations for validation purposes of remote sensing observations. For example, one major argument for the implementation of the Argo observing system is the fact that these data are fundamental for accuracy of remote sensing products - and this is valid also nowadays.

Lines 775 to 783: Readers might expect a stronger outcome / summary? Builds on large communities for each component, needs more or higher level of coordination? What made it different from (improved over) prior efforts? Here, EEI calculated through 2018 (with some extrapolations for some components since 2016). How fast does the system evolve? How fast do our observations evolve? Next inventory in 5 years? Two years? What does the science require, compared to what the observations can provide? Vast amount of work summarized here but the conclusion leaves readers wondering what next? Numerous small typos that I hope the proofreaders will correct. We specifically would like to that the reviewer for this comment, and we fully agree with the reviewers comment. Accordingly, we have applied major revision to the last paragraph of the conclusion, and we thus hope that we could properly reply to the reviewers' request and concern.

%%%%%%%%%%%%%%%%%%%%%%%%%%%%%%%%%%%%%%%%%%%%%%%%%%
Reviewer 3 John Fasullo, fasullo@ucar.edu

[Figure]
Summary The manuscript provides a WRCP-SPARC group summary of the status of our understanding of Earth's Energy Imbalance since 1960 and covers much of the recent progress in documenting the component level evolution of energy uptake, including ocean, atmosphere, cryosphere, and land. An effort is made to quantify some of the uncertainties. I find the manuscript to be reasonably well organized and clear in scope.

I do have numerous suggestions to improve the writing, which is in instances awkward, but these are generally minor. I also strongly suggest that the tables be reconsidered in favor of a graphical summary such as the one below, which I believe will be much more effective at conveying the key points. I also think additional context should be added, particularly to the section on boreholes, to address the limitations of this data source that I believe go largely unmentioned. I have similar concerns regarding the ocean heat content section. Lastly I also feel some effort should be made to minimize the aliasing of internal variability onto the trends. We thank the reviewer for the overall comments, which helps to improve the manuscript, and we provide a point-by-point reply below.

Major Issues

1) The tables are largely ineffectual at conveying the key points. See suggestions below. We specifically like to thank the reviewer for this comment, and we thus provide a new figure following the reviewers recommendation.

2) Basic characteristics of the borehole data go unmentioned and associated caveats in forming an estimate of global-land heat uptake go unaddressed. We have included a new paragraph on the text discussing the strengths and limitations in borehole recon-structions as suggested by Dr. Fasullo.

3) The role of sampling uncertainty in the ocean heat content estimates gets swept

under the rug when the multi-estimate average is performed and Figure 1/2 and their discussion gives an overly confident depiction in my view. This issue should be raised and emphasized in my opinion - as it is a major caveat to the summaries. We have now included a specific discussion on the uncertainties.

4) No effort is made to account for internal variability in accounting for trends. This is particularly relevant for the atmosphere where the role of ENSO is strong. I suggest some effort be made and a number of reasonable approaches exist. We agree with the reviewer on this comment, and we have added further discussion and references in the text for this topic. Moreover, we have used now a weighted least square fit taking into account the error bars.

Minor Issues

78: "internal climate modes can temporarily alter the energy balance for periods of sub-monthly to several decades"; deserves several references Thank you for spotting this, and we have added the AR5 reference as the specific box provides a comprehensive list of references and the highly relevant CMIP5 model study of Palmer and McNeall (2014). We have amended the text to refer to "sub-annual to multi-decadal timescales", which is more consistent with the focus of the timeseries presented in the manuscript.

92-93: this statement also deserves several references - e.g. Cheng et al EOS Thank you, we have added the reference.

101: there is substantial noise in this equation and this should be acknowledged with an additional term, e.g. epsilon, even in a conceptual model since it is unavoidable. Yes, this is a good point. We have added a sentence to this paragraph to acknowledge the complication of internal climate variability and cited the work of Palmer & McNeall (2014). We have also moved some relevant text from the following paragraph here, which discusses ENSO-related variability in EEI as seen in the CERES observations of Loeb et al (2012). For simplicity, and to emphasise the fundamental relationships between forcing, feedback and EEI, we prefer to keep the equation as it is.

[Figure]

Interactive
comment

132: Table 1 would be more useful as a figure such as in the ocean heat uptake summary below as it provides a visual for both period and magnitude/uncertainty: ï£://www.realclimate.org/images//oh_uptake2.png A specific thank you for this proposition, and we have followed the recommendation accordingly.

183: I find "change in behavior" to be vague and unclear. Agreed, and removed.

187: I find "Implications of ocean warming: : :, and have in turn impacted the ocean itself" to be both unclear and poorly related to the following sentence where the impacts on the ocean go unmentioned and thus are not examples. Agreed, and removed.

194: "challenges of Ocean Heat Content (OHC) estimates" is poor grammar. what I think is mean is "opportunities and challenges in forming OHC estimates: : :". Agreed, and changed.

196. Remove "Early" since before 1900 is already stated. Agreed, and changed. 215: "further challenged the global scale ocean heat content estimate" perhaps should read "posed a further challenge for the global scale estimation of ocean heat content" Agreed, and changed.

216: "correct biases" Agreed, and changed.

222: "with satellite-derived: : :" Done.

233: "The opportunity for improved OHC estimation provided by Argo is tremendous" Thank you, changed.

239: "allow further observing system recommendations" could read "inform further observing system recommendations" Thank you, done.

258: 2018). Done.

265: What initiative? Thanks, we have changed to 'This initiative' to clarify.

268: We do (present tense?) Yes, thank you, and changed accordingly.

271: Is an "ensemble mean and standard deviation of near-global " an improvement? The assumption is that all errors are random and not systematic. But are they? I think the case would need to be made, and I don't think it is largely true. Thank you, and we fully agree. Accordingly, we have added the following sentence: 'The ensemble mean approach does not provide an improved uncertainty evaluation, but indicates the agreement of the different products used in this study.'

273: "is needed to include all missing in situ-based products" is not clear what is being referred to? what are the main data sources that have yet to be included (in reanalyses for example)? Figure 1: What exactly is in the sigma? XBT drop rate uncertainties? sampling uncertainties? I sense that the answer is "none" but rather the spread in the derivations of previous products, which themselves have much larger uncertainties and that averaging together does not reduce. This is a point worth emphasizing as Fig 1 seem overconfident as-is. I thing that a true "uncertainty" bound be shown somehow (whiskers?). And I suggest the reference to "values are given for the ocean surface" be made clearer as the figure itself obviously doesn't deal with the ocean surface (just the area used for the heating rates). Figure 2: As with Figure 1 there is an underlying sampling/data uncertainty that is not dealt with but is very relevant to the values plotted and should be shown. Ultimately I think this uncertainty creates an important caveat for the results. Also the figure/caption lack units). We have removed "include all missing in situ-based products". And a more thorough discussion on the uncertainty estimate: method, assumption and caveats is included here. In the final section, we have discussed the next steps for uncertainty estimate, which is a long-standing unresolved issue in the community.

311: But this result is already well known? Yes, we agree, thanks for spotting this, and we have refined the sentence accordingly: . . . , which is in agreement with previous results (e.g. Abraham et al., 2013).

370: Eq (1) seems off as the surface enthalpy fluxes for P and E are already contained in FS, unless what is meant is the very small term of the sensible heat of rainfall? (other

components of enthalpy such as evaporation are already in FS). Given the description later (in lines 388-90) I think sensible heat is the correct term? as enthalpy also entails the latent component. FPE essentially measures the energetic effect of P and E occurring at temperatures different from 0°C. Hence, the evaporation contribution to FPE is the change in the latent heat flux computed with a temperature-dependent latent heat relative to that obtained when using constant latent heat (the latter is contained in FS). The precipitation contribution FPE arises from the temperature difference of rain from a reference temperature (here 0°C). Taking these effects together, FPE arises from the fact that P and E occur at different temperatures in a global average sense. We modified the text to make this point clearer.

381: Why "global" evaporation and precipitation? "Global" is appropriate here as FPE is only unambiguously defined if P and E balance (i.e. if there is no net mass flux), which in general is only satisfied for global averages. Moreover, the equation is written in its global average form and hence "global" does not represent any restriction of its validity.

398: "account for" Done.

406: "neglect" Done.

425: "used" : : : Done.

Figure 3: The caption does not adequately distinguish between the terms plotted in the middle and bottom row. Perhaps it could better emphasize that one is the anomaly while the other is the annual tendency (change and trends can be taken as synonymous). Ok, caption improved accordingly.

476: An important point is raised here. Clearly El Niño is aliasing on to the trends to some extent. Shouldn't an effort be made to remove such effects? Various approaches have been tried in the literature and I'm not advocating for any single one, but to make no effort seems lazy. We have checked against trends fitted to ENSO-corrected AHC

anomalies (with the ENSO signal regressed out via the NINO 3.4 Index); due to the sufficiently long periods of trend-fitting these are quite similar to the trends fitted directly. We have now inserted a sentence pointing to this sensitivity check.

544: There is an odd symbol after "condition."? We have corrected that.

555: The entire discussion is very generic here. What is the spatial sampling of boreholes that can be used and what temporal resolution for GSTHs can they provide and at what spatial sampling / accuracy can they be used. The temporal resolution of borehole reconstructions ranges from decades to centuries, depending on the target of the analysis. For example, Cuesta-Valero et al. (2020) uses 25yr, 40yr and 50yr time steps to reconstruct surface conditions for the last four centuries. In this study, the authors use a global database composed of 1079 profiles, which is proved to be able to capture the evolution of global surface conditions in comparison with CRU data. Furthermore, other studies have shown that the distribution of measured profiles is enough to represent the evolution of global surface conditions (e.g., Beltrami & Bourlon, 2004; Pollack & Smerdon, 2004; González-Rouco et al., 2006). We have included a few lines addressing this point on the text.

How does groundwater corrupt/ influence inferred histories? Indeed, the effect of groundwater on measured borehole profiles can affect the reconstructed climatic signal. Nevertheless, this spurious signal can be detected in a preliminary screening of the data. All borehole profiles in the cited literature were screened to select only the logs suitable for climate reconstructions. That is, those affected by groundwater flows were not used for climate studies. We have added a few lines about that on the text.

What validation of the reconstructions exist and what does all of this reveal that is both new and robust about "the evolution of the climate system". The borehole methodology has been validated in numerous works (González-Rouco et al., 2006; and García-García et al. 2016). Additionally, the work of Cuesta-Valero et al. (2020) compares directly borehole reconstructions with meteorological observations from the CRU

database, showing good agreement. We include now a few lines clarifying this point.

565. How is a global estimate inferred from what must be very discrete observations? We think that the reviewer is asking about the work of Huang (2006). These estimates are based on CRU data, that is, Huang (2006) did not use individual stations but the gridded CRU product.

Table 2: Again I think Table 2 would be best as a figure such as suggested for Table 1. We thank Dr. Fasullo for his suggestion. Nevertheless, we think that the suggested figure can be confusing given the number of estimates to be plotted. Thus, we still think that Table 2 is effective in summarizing the available estimates of continental heat storage and ground heat flux from borehole temperature profiles, meteorological data, and FluxNet observations.

639: Remove "in sea-ice extent" as it is redundant. Due to revision, the sentence had been removed, and thus comment not applicable anymore.

624: "with ice loss" Sentence had been modified due to revision process.

652: "which are based on a modeled surface mass balance combined with: : :" Revision has changed this sentence.

654: "estimate of floating ..." Revision has changed this sentence.

740: "needs further evaluation" Revision has changed this sentence.

761: There is oddly no mention of the need to develop modeling/assimilation capabilities as a priority. The authors may view these as distinct from the observational imperatives however in my view they are not, as they are central to the generation of ocean/atmosphere/cryosphere models that are vital for improving the reanalyses used. The same can be argued for terrestrial system models given the poor coverage of borehole measurements. Thank you for this suggestion. We had included an additional paragraph that discusses the link between observations and improved modelling systems - and how these can also help us better understand and prioritize future

observations.

References Beltrami, H., & Bourlon, E. (2004). Ground warming patterns in the Northern Hemisphere during the last five centuries, Earth Planet. Sc. Lett., 227, 169–177.

García-García A., Cuesta-Valero F.J., Beltrami H. and Smerdon J.E. (2016). Simulation of Air and Ground Temperatures in PMIP3/CMIP5 Last Millennium Simulation: Implications for Climate Reconstructions from Borehole Temperature Profiles. Environmental Research Letters, 11(4):044022, doi:10.1088/1748- 9326/11/4/044022.

González‐Rouco, J. F., Beltrami, H., Zorita, E., and von Storch, H. (2006), Simulation and inversion of borehole temperature profiles in surrogate climates: Spatial distribution and surface coupling, Geophys. Res. Lett., 33, L01703, doi:10.1029/2005GL024693. Pollack, H. N., & Smerdon, J. E. (2004), Borehole climate reconstructions: Spatial structure and hemispheric averages, J. Geophys. Res., 109, D11106, doi:10.1029/2003JD004163.

---

## Editor Decision (ED1)

Line 331: "neglecting shallow waters can account for 5-10%" of what. Of total OHC? Of only the 0 - 300 m OHC? Clarification needed here. In the following lines (332, 333), the authors specify 0 - 2000 m underestimated by 10% due to latitudinal constraints. We need similar specificity with respect shallow bathymetric limits. Good section, just needs a bit of clarification.

Line 342: replace "All-time" with 'All time'?

Line 351: CAR2009 in the legend but CARS2009 in the figure (and in the URL). An error in the legend?

Line 551, Figure 3: Important figure but still hard to read. Does not scale / zoom. Authors can fix this during proofreading.

Line 562 and following: Section 3 Land uses 1.5 line spacing, different to other sections. Authors can correct these differences after typesetting and during proofreading.

Line 564: Something like this opening sentence should also have preceded the atmospheric section (Section 2). If true of land, certainly true of atmosphere.

Line 565: land-based rather than land based?

Line 584: "small, but persistent" remove the comma?

Line 621: proofreaders will question capitalization of LANE 1923 citation but it at least seems consistent with reference list. Artifact carried forward from bibliographic software?

Lines 653 - 658: These sentences seem redundant with previous section? If authors add a section on borehole climatology, which I agree adds substantial value, then they do not need to re-introduce the topic here?

Line 805: comparably rather than comparable?

Line 834: Figure 7 copied from elsewhere? We need a complete high-resolution (scalable) version here. Make the upgrade during proofreading?

Line 847: Something awkward here? Remove the comma?

Line 919 and Figure 9: If authors present atmospheric $CO_2$ reductions as a rate, they need a time unit. The text narrative seems clear. Perhaps the term 'rate' causes the confusion? -57 ppm represents a cumulative $CO_2$ accumulation/removal, not a rate.

One can check this number and the underlying assumptions. The authors, working from energy imbalance, estimate 57 ppm reduction (410 ppm back to 353 ppm)

needed to achieve 0.87 W/m2 increase in outgoing energy. E.g. so that outgoing increases to match incoming. Eyeball from Moana Loa $CO_2$ curve, planet last had global average of 355 ppm $CO_2$ in 1990. From Global Carbon Budget, atmospheric growth of $CO_2$ from 1990 through 2018 sums to 117 GtC, or 55 ppm. Given uncertainties all around (including in my eyeball estimates), restoration (removal) requirements calculated from energy imbalance, -57 ppm, match very closely what carbon budget shows as incremental $CO_2$ inputs, 55 ppm, over roughly the past 30 years. Two very different global budget approaches, one from radiative (energy/heat) viewpoint and second from carbon emissions estimates, arrive at the same answer? To first order, acknowledging various necessary assumptions and uncertainties, we quantify with confidence human impact on climate? Small effort on my part to make this comparison, in part because at this point I hold good knowledge of both products. Worth including something like this as a non-radiative non-energy confirmation?
.
Line 972: For cryosphere, we don't need to measure gravity itself but rather use gravimetric measurements to constrain ice mass change and water redistribution?

Line 1008: at the forward or leading edge, not at the (peripheral) edge?

Please take great care to reference primary peer-reviewed source literature when possible. I react mostly to WMO 2020, their 2019 annual statement on climate. I understand benefits: annual, recent, carries WMO imprimatur, etc. But those WMO reports remain highly derivative, second-hand at best. They compile entirely from outside sources, mostly from NOAA. Internal staff compile and write them, with only weak internal review, from almost exclusively a meteorological viewpoint. Unlike IPCC reports, WMO annual reports get virtually no community review. If you need an annual product, consider the AMS annual statement on climate which incorporates a broader range of inputs and gets strong review. (https://www.ametsoc.org/index.cfm/ams/ publications/bulletin-of-the-american-meteorological-society-bams/state-of-the-climate/) Often WMO hurries their annual report, even issuing provisional versions before they have a full year of data, to promote themselves within UN system. AMS follows a more strict and more regular assembly and review schedule. Not a big issue and you have referenced the WMO report correctly; AMS report would add more credibility to your product. I think it has all or most of the same information. My caution but your choice.

---

## Author Response (AR2)

**Point-by-point reply**

Line 331: "neglecting shallow waters can account for 5-10%" of what. Of total OHC? Of only the 0 - 300 m OHC? Clarification needed here. In the following lines (332, 333), the authors specify 0 - 2000 m underestimated by 10% due to latitudinal constraints. We need similar specificity with respect shallow bathymetric limits. Good section, just needs a bit of clarification.
Yes, thank you for rising this, and 0-2000m depth OHC trends have been added to the shallow water part as well.

Line 342: replace "All-time" with 'All time'?
Thank you, done.

Line 351: CAR2009 in the legend but CARS2009 in the figure (and in the URL). An error in the legend?
Thank you, an error in the figure caption, done.

Line 551, Figure 3: Important figure but still hard to read. Does not scale / zoom. Authors can fix this during proofreading.
Yes, we have already prepared a high-quality version of this figure (we have this for all figures), and can provide this during the proof read process?

Line 562 and following: Section 3 Land uses 1.5 line spacing, different to other sections. Authors can correct these differences after typesetting and during proofreading.
Yes, thank you, and I assume this will be solved during the processing? It is fixed now for this resubmission.

Line 564: Something like this opening sentence should also have preceded the atmospheric section (Section 2). If true of land, certainly true of atmosphere.
Thank you, and we agree. We have modified the first sentence by: 'While the amount of heat accumulated in the atmosphere is small compared to the ocean, warming of the Earth's near-surface air and atmosphere aloft is a very prominent effect of climate change, which directly affects society.' and the last sentence of the second paragraph by 'In contrast, long-term heat accumulation in the atmosphere is limited by its small heat capacity as the gaseous component of the Earth system (von Schuckmann et al., 2016).'

Line 565: land-based rather than land based?
Yes, thank you, done.

Line 584: "small, but persistent" remove the comma?
Yes, thank you, done.

Line 621: proofreaders will question capitalization of LANE 1923 citation but it at least seems consistent with reference list. Artifact carried forward from bibliographic software?
Yes, thank you, done.

Lines 653 - 658: These sentences seem redundant with previous section? If authors add a section on borehole climatology, which I agree adds substantial value, then they do not need to re-introduce the topic here?

We checked the lines and we think there is no repetition. We describe the borehole climatology in the first part of the section, and then we focus on the previous estimates of continental heat content, from boreholes and meteorological observations. So we think there is no need to change the text

Line 805: comparably rather than comparable?
Yes, thank you, done.

Line 834: Figure 7 copied from elsewhere? We need a complete high-resolution (scalable) version here. Make the upgrade during proofreading?
Yes, we have a high-resolution version available, even .ai (done by a graphic designer).

Line 847: Something awkward here? Remove the comma?
Yes, thank you, and the sentence is modified to' Over the last quarter of a decade this Earth heat inventory reports - in agreement with previous publications - an increased rate of Earth heat uptake reaching up to 0.9 W/m$^2$ (Fig. 7).'

Line 919 and Figure 9: If authors present atmospheric CO2 reductions as a rate, they need a time unit. The text narrative seems clear. Perhaps the term 'rate' causes the confusion? -57 ppm represents a cumulative CO2 accumulation/removal, not a rate. One can check this number and the underlying assumptions. The authors, working from energy imbalance, estimate 57 ppm reduction (410 ppm back to 353 ppm) needed to achieve 0.87 W/m2 increase in outgoing energy. E.g. so that outgoing increases to match incoming. Eyeball from Moana Loa CO2 curve, planet last had global average of 355 ppm CO2 in 1990. From Global Carbon Budget, atmospheric growth of CO2 from 1990 through 2018 sums to 117 GtC, or 55 ppm. Given uncertainties all around (including in my eyeball estimates), restoration (removal) requirements calculated from energy imbalance, -57 ppm, match very closely what carbon budget shows as incremental CO2 inputs, 55 ppm, over roughly the past 30 years. Two very different global budget approaches, one from radiative (energy/heat) viewpoint and second from carbon emissions estimates, arrive at the same answer? To first order, acknowledging various necessary assumptions and uncertainties, we quantify with confidence human impact on climate? Small effort on my part to make this comparison, in part because at this point I hold good knowledge of both products. Worth including something like this as a non-radiative non-energy confirmation?
Thank you for this comment. We have discussed your proposition with some co-authors, and we thus propose at this stage only minor changes, ie: we have removed the wording 'rates' from the caption which we agree was rather confusing, and we have added the sentence: 'Atmospheric CO2 was last 350 ppm in the year 1988, and global Earth's surface temperature was then +0.5°C relative to the pre-industrial period (relative to the 1880-1920 mean) (Hansen et al., 2017; Friedlingstein et al., 2019).'

Line 972: For cryosphere, we don't need to measure gravity itself but rather use gravimetric measurements to constrain ice mass change and water redistribution?
Thank you and we have changed to gravimetric measurements.

Line 1008: at the forward or leading edge, not at the (peripheral) edge?
Thank you, yes, and applied.

Please take great care to reference primary peer-reviewed source literature when possible. I react mostly to WMO 2020, their 2019 annual statement on climate. I understand benefits:

annual, recent, carries WMO imprimatur, etc. But those WMO reports remain highly derivative, second-hand at best. They compile entirely from outside sources, mostly from NOAA. Internal staff compile and write them, with only weak internal review, from almost exclusively a meteorological viewpoint. Unlike IPCC reports, WMO annual reports get virtually no community review. If you need an annual product, consider the AMS annual statement on climate which incorporates a broader range of inputs and gets strong review.
(https://www.ametsoc.org/index.cfm/ams/ publications/bulletin-of-the-american-meteorological-society-bams/state-of-the-climate/) Often WMO hurries their annual report, even issuing provisional versions before they have a full year of data, to promote themselves within UN system. AMS follows a stricter and more regular assembly and review schedule.

Not a big issue and you have referenced the WMO report correctly; AMS report would add more credibility to your product. I think it has all or most of the same information. My caution but your choice.

Thanks a lot, and I have added to each WMO reference another one, such as BAMS, or Richter-Menge, J., M.L. Druckenmiller, and M. Jeffries, Eds., 2019: Arctic Report Card, 2019. https://www.arctic.noaa.gov/Report-Card.
And for the $CO_2$ concentrations I have started with the last ESSD value, yes, you are completely right…